# Discovery of a potent, Kv7.3-selective potassium channel opener from a Polynesian traditional botanical anticonvulsant
Geoffrey W. Abbott ✉ & Rían W. Manville

Plants remain an important source of biologically active small molecules with high therapeutic potential. The voltage-gated potassium (Kv) channel formed by Kv7.2/3 (KCNQ2/3) heteromers is a major target for anticonvulsant drug development. Here, we screened 1444 extracts primarily from plants collected in California and the US Virgin Islands, for their ability to activate Kv7.2/3 but not inhibit Kv1.3, to select against tannic acid being the active component. We validated the 7 strongest hits, identified *Thespesia populnea* (*miro, milo*, portia tree) as the most promising, then discovered its primary active metabolite to be gentisic acid (GA). GA highly potently activated Kv7.2/3 (EC$_{50}$, 2.8 nM). GA is, uniquely to our knowledge, 100% selective for Kv7.3 versus other Kv7 homomers; it requires S5 residue Kv7.3-W265 for Kv7.2/3 activation, and it ameliorates pentylenetetrazole-induced seizures in mice. Structure-activity studies revealed that the FDA-approved vasoprotective drug calcium dobesilate, a GA analog, is a previously unrecognized Kv7.2/3 channel opener. Also an active aspirin metabolite, GA provides a molecular rationale for the use of *T. populnea* as an anticonvulsant in Polynesian indigenous medicine and presents novel pharmacological prospects for potent, isoform-selective, therapeutic Kv7 channel activation.

Botanical extracts have been used as medicines by human species since the Paleolithic period, and the practice continues worldwide today[1,2]. Our recent work has uncovered a prominent role for voltage-gated potassium (Kv) channels as targets for metabolites found in botanical medicines, in some cases helping to explain their therapeutic actions[3–5]. Members of the Kv7 (KCNQ) family of Kv channels are particularly highly regulated by plant metabolites[6], which is significant given the prominent role of Kv7 channels in the physiology of both excitable and non-excitable cell types[7,8], and the plethora of Kv7 gene sequence variants linked to human diseases[9,10].

In neurons, heteromeric channels formed by Kv7.2 and Kv7.3 (Kv7.2/3) have a strong influence on cellular excitability and are the primary molecular correlate of the muscarinic acetylcholine receptor-inhibited "M-current"[11]. Heteromeric Kv7.2/3 channels are expressed in the axon initial segment, where they act as gatekeepers for neuronal signaling, as well as in the soma[12]. Loss-of-function mutations in the *KCNQ2* gene that encodes Kv7.2 cause KCNQ2 developmental epileptic encephalopathy (KCNQ2-DEE), a severe disorder that causes seizures and often profound developmental delay[13–15]. Similarly, KCNQ3 and KCNQ5 loss-of-function (and some gain-of-function) mutations cause developmental delay and/or seizures[16–20]. The severity of diseases such as KCNQ2-DEE, and the efficacy of the first-in-class neuronal Kv7 channel opener (retigabine; see below) in this and other seizure disorders have resulted in Kv7.2/3 being a major target for the development of novel anticonvulsants, but there have been difficulties in safe targeting of this channel, as described below. Moreover, epilepsy is a relatively common neurological disorder, with 1.1% of the US adult population (almost 3 million adults) having active epilepsy and almost half a million children[21,22]. Importantly, in about a third of people with epilepsy, the condition is intractable, meaning that new therapies are highly sought-after[23].

The first-in-class anticonvulsant retigabine (ezogabine) works by binding to Kv7.2/3 channels to negative-shift their voltage dependence of activation, making them open more easily at hyperpolarized potentials. This reduces excitability of neurons expressing Kv7.2/3, making them less likely

Bioelectricity Laboratory, Department of Physiology and Biophysics, School of Medicine, University of California, Irvine, CA, USA.
✉ e-mail: abbottg@hs.uci.edu

to participate in the abnormal excitability and excessive firing associated with seizure activity[24,25]. Unfortunately, retigabine was voluntarily withdrawn from clinical use in 2017 due to limited use after restrictions because of side effects unrelated to its action on Kv7.2/3[26,27]. New synthetic openers are under commercial development because of the suitability of neuronal Kv7.2/3 as an anticonvulsant target. Among the challenges for Kv7.2/3 opener development are specificity (avoiding action on closely related Kv7 and other Kv channels) and potency (lower required doses combined with selectivity can reduce likelihood of side effects).

Here, we took an alternative approach to finding new Kv7.2/3 openers. We screened plant extracts, primarily from vascular plants collected in California and the United States Virgin Islands, for Kv7.2/3 opening ability. We and others previously found that tannic acid, which is present in many plants, opens Kv7.2/7.3 channels. While this is of interest, tannic acid does not cross the blood-brain barrier efficiently[28] and can cause gastrointestinal side effects[29]; in addition, we wished to discover novel Kv7.2/7.3 openers in the present study. We recently found that tannic acid inhibits Kv1.3, a potassium channel expressed in and required for activation of T cells[30]. Kv1.3 inhibition is, therefore, a useful property for anti-inflammatory agents[31], but here we used Kv1.3 inhibition as a novel method for screening out plant extracts containing substantial quantities of tannic acid, so that we would have a better likelihood of discovering non-tannic-acid Kv7.2/3 openers. Our unbiased screen indeed identified a plant reportedly used as an anticonvulsant in Polynesian traditional medicine as the top hit[32], from which we discovered what is to our knowledge the most potent and selective Kv7.3 opener reported to date.

## Results

### A subset of plant extracts open Kv7.2/3 while inhibiting <25% Kv1.3 current

High-throughput screening of 1444 plant extracts (1:50 dilution; equivalent to 5 mg fresh plant matter starting material/ml) for Kv7.2/3 opening revealed many extracts with activity on one or both channels (Fig. 1a). In a recent study[30] we focused on the extracts that produced the greatest inhibition of Kv1.3 ≥ 95% (at 1:50 dilution) and also increased thallium flux in the Kv7.2/3 FLIPR assay by ≥50%. This produced a subset of 15 extracts, and we found that tannic acid, present in many of the plants in the subset, both opened Kv7.2/3 and inhibited Kv1.3. Tannic acid is effective as a topical anticonvulsant/analgesic but does not cross the blood-brain barrier efficiently (although its metabolites do)[28] and can cause gastrointestinal side effects[29] so here we wanted to identify other Kv7.2/3 openers. In the present study, to identify potential anticonvulsants, we, therefore, focused first on plant extracts that increased Kv7.2/3 thallium flux >75% (a value chosen because it encompassed 50–100% of the maximal Kv7.2/3 opening effects we observed in the extracts, yet included only the top 0.5% of extracts) but inhibited Kv1.3 < 95% (a value chosen because in our recent study we investigated extracts showing ≥95% Kv1.3 inhibition and concluded that the major molecular correlate of this effect was tannic acid[30] (Fig. 1a). Using this approach, we first identified 7 extracts that increased Kv7.2/3 thallium flux by 76–150% with varied degrees of Kv1.3 inhibition, between ~0 and 93% (Fig. 1a, b).

Two extracts among the top 7, from portia tree (*Thespesia populnea*), collected from a beach in St. John, USVI and fourwing saltbush (*Atriplex canescens*), collected from Mojave National Preserve, CA, had repeat samples (shown in gray text), collected from different locations, that had similar properties but fell just outside of our criteria for this subscreen, indicating consistency of effects between same-species samples (Fig. 1a, b). There were two distinct groups among the top 7 hits. The first contained those exhibiting 75–93% Kv1.3 inhibition—fourwing saltbush and a suspected hybrid between sugar bush (*Rhus ovata*) and a garden escape species, possibly golden wattle (*Acacia pycnantha*), collected from a dry riverbed in Santa Monica Mountains, close to Malibu, CA. The second group each exhibited <25% Kv1.3 inhibition and comprised 5 species—portia tree; one-seeded pussy-paws (*Calyptridium monospermum*) and a species of stickseed (*Hackelia* sp.) from Yosemite National Park, CA; bullwhip kelp (*Nereocystis luetkeana*—an

edible seaweed, therefore an alga, not a plant) from northern CA; and Spanish needle (*Palafoxia arida*) from Boyd Deep Canyon, CA (Fig. 1a, b).

### Hit validation reveals portia tree as the most promising anticonvulsant-containing extract

We validated effects of the top 7 hits (Fig. 2a) on Kv7.2/3 using manual two-electrode voltage clamp electrophysiology (TEVC) (Fig. 2b), shown in order of most (upper) to least (lower) Kv1.3 inhibitory effects from the screen (see Fig. 1). Effects of Kv7.2/3 openers are typically difficult to fully assess by examining currents elicited during the prepulse family of depolarizations (Fig. 2c) because opening action often involves a negative shift in the voltage dependence of activation, most prominent at physiologically important membrane potentials, at which the driving force and therefore current is relatively low. Thus, tail currents are recorded in which channels are opened at the prepulse potentials and then measured at a potential that gives a higher driving force but is not so depolarized to open the channels too quickly, in this case, −30 mV (Fig. 2d, e). Tail current quantification revealed that the sugarbush hybrid had an effect consistent with known tannic acid effects[29,33], i.e., large increase of current at hyperpolarized potentials but inhibition at depolarized potentials (Fig. 2d); this was consistent with the placing of this sample at 93% Kv1.3 inhibition in the original screen (Fig. 1a). Thus, while the sugarbush hybrid sample increased constitutive current and hyperpolarized resting membrane potential ($E_M$) in oocytes expressing Kv7.2/3 (Fig. 2e, f), it would likely not be optimal for anticonvulsant development due to tannic acid being the active component. Saltbush gave interesting effects, including hyperpolarization of voltage dependence of activation ($V_{0.5act}$) and of $E_M$, but it also caused an inward current at depolarized voltages that was potassium-nonselective (it reversed polarity at 0 mV) (Fig. 2a–f). We attribute this to saponins known to be present in *Atriplex* sp[34]. causing membrane leak, and therefore we also ruled out this extract for further study as an anticonvulsant. In contrast, portia tree extract (red boxed region) negative-shifted the voltage dependence of activation by −12.8 ± 1.2 mV (Fig. 2b–e) and the $E_M$ by −7.7 ± 1.6 mV (Fig. 2f) and caused neither artifacts nor locked open current at −80 mV that would indicate tannic acid contribution, suggesting it as a strong candidate for further study. Of the remaining samples, only the stickseeds extract marginally negative-shifted $V_{0.5act}$, without causing a statistically significant hyperpolarization of $E_M$ (Fig. 2a–f).

Portia tree, which we collected from a beach in St. John, USVI, is pantropically distributed in coastal regions from India to the Caribbean, has been used in various traditional medicinal practices around the world, including as an anticonvulsant[32], and its major metabolites previously identified[35]. We therefore next studied the effects of its major metabolites by composition on Kv7.2/3 activity.

### TEVC screening of portia tree compounds reveals gentisic acid as a Kv7.2/3 opener

We tested the effects of 12 of the most abundant compounds present in portia tree extract[35], using TEVC. At 100 μM, 2 of the 12 compounds shifted Kv7.2/3 $V_{0.5}$ act; gentisic acid (GA) hyperpolarized $V_{05act}$ by −11.5 ± 1.0 mV ($n = 10$) (Fig. 3a) and $E_M$ by −8.8 ± 2.2 mV ($n = 10$, $p = 0.0008$) mV (Fig. 3b). The structurally related protocatechuic acid (PA) hyperpolarized $V_{05act}$ by −6.9 ± 0.5 mV ($n = 7$) (Fig. 3c) and $E_M$ by −3.6 ± 1.5 mV ($n = 7$, $p = 0.03$) (Fig. 3d). Each compound also increased peak tail current by up to 50% (Fig. 3a, c). The remaining compounds altered neither $V_{0.5act}$ nor $E_M$ (Fig. 3a–d).

### Gentisic acid is Kv7.3-selective

As GA and PA acid each increased Kv7.2/3 activity and negative shifted $V_{0.5act}$, we tested effects of adding them both together to Kv7.2/3 (each at 50 μM), but the $V_{0.5act}$ shift was not different from that of GA alone (100 μM) at −11.24 ± 4.5 mV (Fig. 4a, b), and the shift in $E_M$ was less than for GA alone (Fig. 4c). As GA was the more efficacious of the two, we completed a GA dose response (Fig. 4d, e) and found that it is also highly potent; the GA EC50 for increasing Kv7.2/3 current at −60 mV was

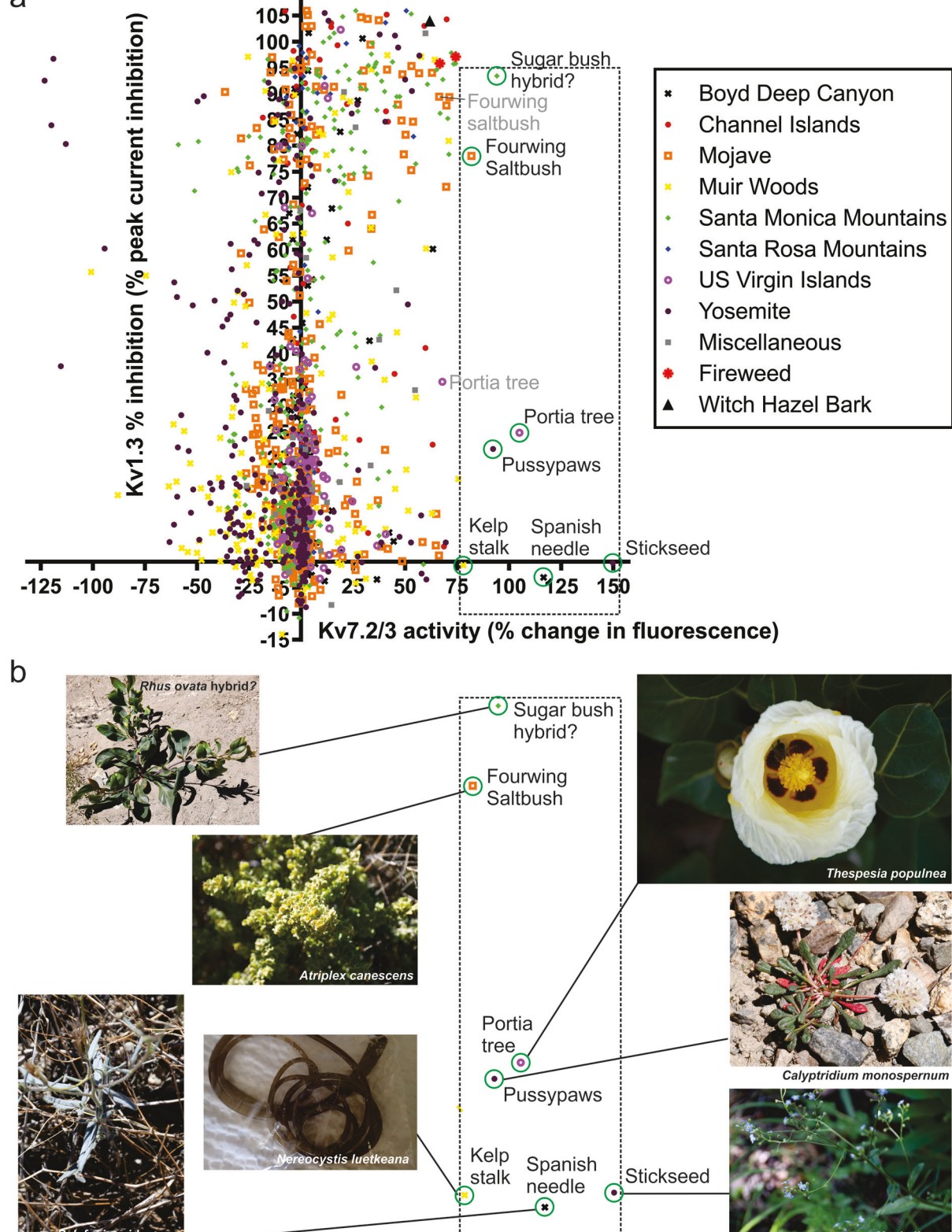

**Fig. 1 | High-throughput screening of plant extracts reveals a subset favoring Kv7.2/3 opening over Kv1.3 inhibition.** Results of 1444-extract screen of plant extract activity against Kv1.3 and Kv7.2/3. Each point indicates the screening result as the mean of a technical triplicate for an individual plant extract (1:50 extract dilution, equivalent to 5 mg fresh plant matter starting material/ml). Adapted from a figure in a previous study by our group[30]. **b** Closeup of dashed box region from panel (**a**) showing identities of plant extracts inducing highest apparent increase in thallium flux in the Kv7.2/3 screen.

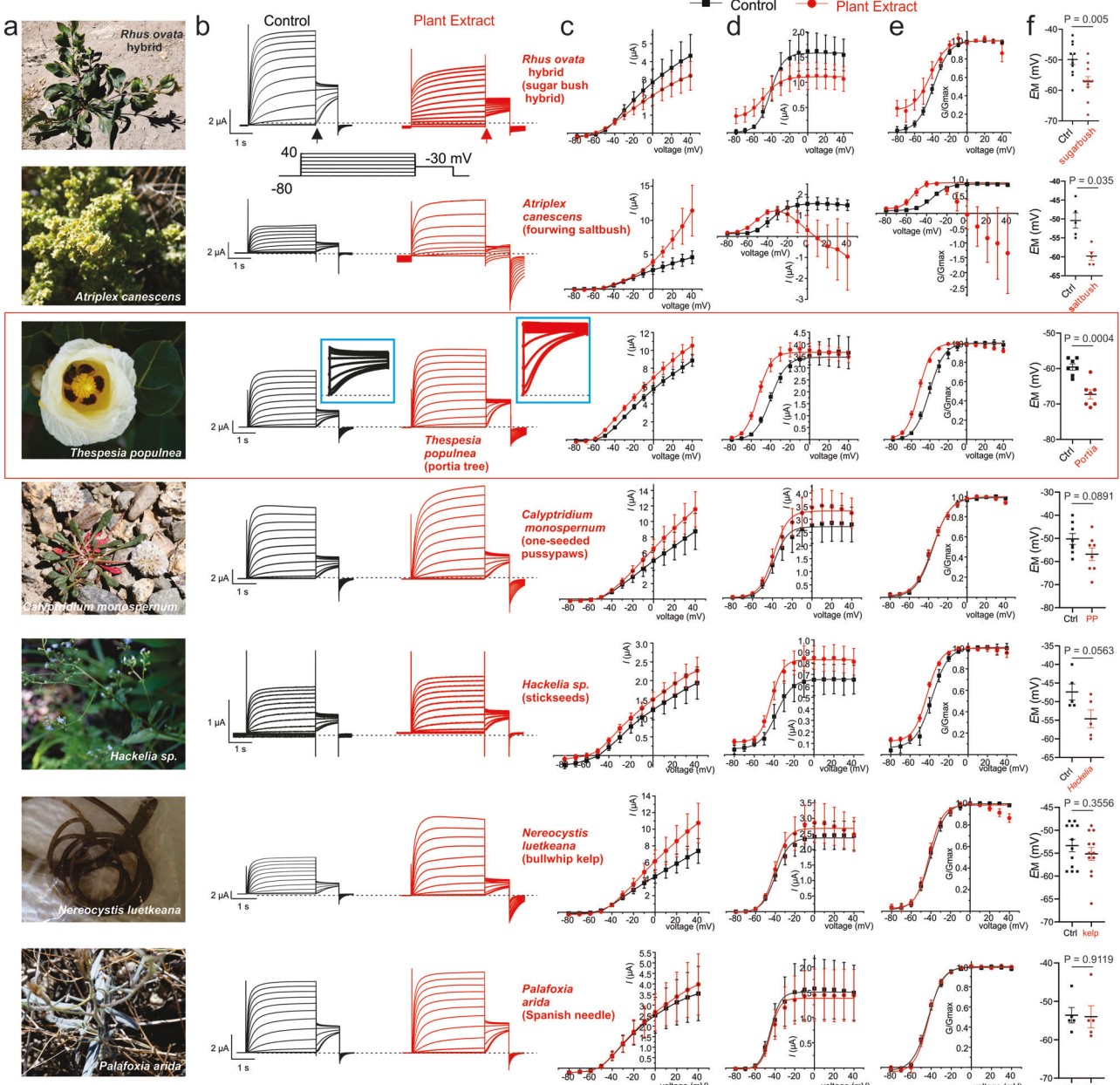

**Fig. 2 | Portia tree extract is the most promising Kv7.2/3 opener among the 7 extract hits. a** Images of plants and alga from which the top 7 hits were extracted. **b** Exemplar Kv7.2/3 traces recorded from oocytes in the absence (Control) or presence of the extracts indicated at 1:50 dilution (5 mg fresh plant matter starting material/ml) using the voltage protocol indicated (inset). Scale bars lower left for each pair of traces; *n* = 5 (stickseeds, saltbush, Spanish needle), 7 (portia tree), 9 (pussypaws), 11 (sugar bush hybrid), 12 (kelp) biologically independent oocytes per group. **c** Mean peak prepulse current for traces as in (**b**); *n* values as in (**b**), in the absence (black) or presence (red) of plant extracts shown on left. **d** Mean peak tail current for traces as in (**b**) (measured at arrows shown in **b**); *n* values as in (**b**), in the absence (black) or presence (red) of plant extracts shown on left. **e** Mean normalized tail current (G/Gmax) for traces as in (**b**) (measured at arrows shown in **b**); *n* values as in (**b**), in the absence (black) or presence (red) of plant extracts shown on left. **f** Mean unclamped oocyte membrane potential for Kv7.2/3-expressing oocytes as in (**b**); *n* values as in (**b**). Error bars indicate SEM. *n* indicates number of biologically independent oocytes. Statistical comparisons by *t*-test. Dashed lines indicate zero current level. Blue boxes contain tail currents for the corresponding traces expanded to preserve scale (test vs control).

15.6 ± 10.3 nM (Fig. 4f), while the EC$_{50}$ values for shifting V$_{0.5act}$ and $E_M$ of Kv7.2/3 were 2.8 ± 4.1 and 2.4 ± 3.7 nM GA, respectively (Fig. 4g).

Strikingly, GA had no effect on Kv7.2 homomers, in terms of current magnitude or V$_{0.5act}$ (Fig. 4h, i, upper); similarly, neither did PA (Fig. 4h, i, lower). In contrast, GA opened Kv7.3* (Kv7.3 containing a A315T mutation to facilitate robust current from the homomer[36]) even at 10 nM (Fig. 4j), and induced an up to −5.9 ± 1.9 mV negative shift in Kv7.3* V$_{0.5 act}$ with an EC$_{50}$ of 0.79 ± 2.0 nM GA (Fig. 4j, k), the maximal recorded shift in Kv7.3* V$_{0.5 act}$ being at 10 nM (Fig. 4l).

Given that GA was highly selective for Kv7.3 over Kv7.2, we investigated the effects of GA on other channels, including heteromeric Kv7 channels found in the brain[37]. At 100 μM, GA had no effect on Kv7.1 and Kv7.4 homomers, or Kv7.2/5 heteromers. GA induced relatively small shifts in the V$_{0.5act}$ of known Kv7.3-containing heteromers Kv7.2/3/5 (−5.3 ± 1.4 mV, *n* = 7) and Kv7.3/5 (−5.5 ± 0.6 mV, *n* = 6) (Fig. 5a–c). Of all the above, GA only shifted $E_M$ in oocytes expressing Kv7.2/3/5, by −6 mV (*p* = 0.0024, *n* = 7) (Fig. 5c). Similarly, GA (100 μM) had no effect on constitutively active channels formed by Kv7.1

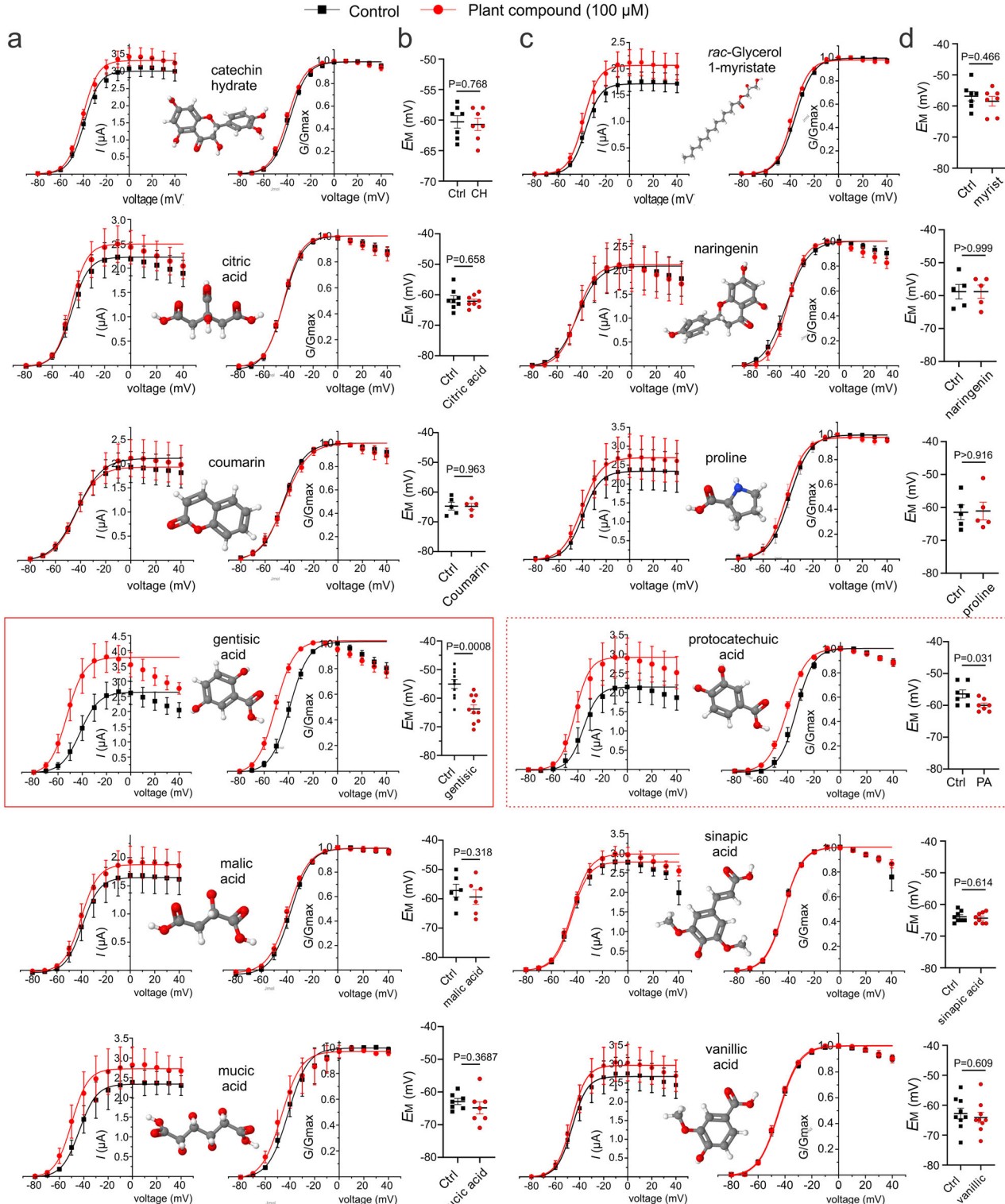

**Fig. 3 | Screening of portia tree metabolites reveals gentisic acid as the principal Kv7.2/3-activating compound. a** Mean peak tail current (left) and normalized tail current (G/Gmax) (right) for Kv7.2/3 channels expressed in oocytes, in the absence (black) or presence (red) of portia tree metabolites indicated (100 μM); $n = 6$ (coumarin, malic acid), 7 (catechin hydrate, mucic acid), 8 (citric acid), 10 (gentisic acid) per group. **b** Mean unclamped oocyte membrane potential for Kv7.2/3-expressing oocytes as in (**a**); $n$ values as in panel (**a**) except coumarin ($n = 5$). CH catechin hydrate. **c** Mean peak tail current (left) and normalized tail current (G/Gmax) (right) for Kv7.2/3 channels expressed in oocytes, in the absence (black) or presence (red) of portia tree metabolites indicated (100 μM); $n = 5$ (naringenin, proline), 7 (protocatechuic acid, *rac*-Glycerol 1-myristate), 8 (sinopic acid), 9 (vanillic acid) per group. **d** Mean unclamped oocyte membrane potential for Kv7.2/3-expressing oocytes as in (**c**); $n$ values as in panel (**c**). PA protocatechuic acid. Error bars indicate SEM. $n$ indicates number of biologically independent oocytes. Statistical comparisons by *t*-test. Black dashed lines indicate zero current level.

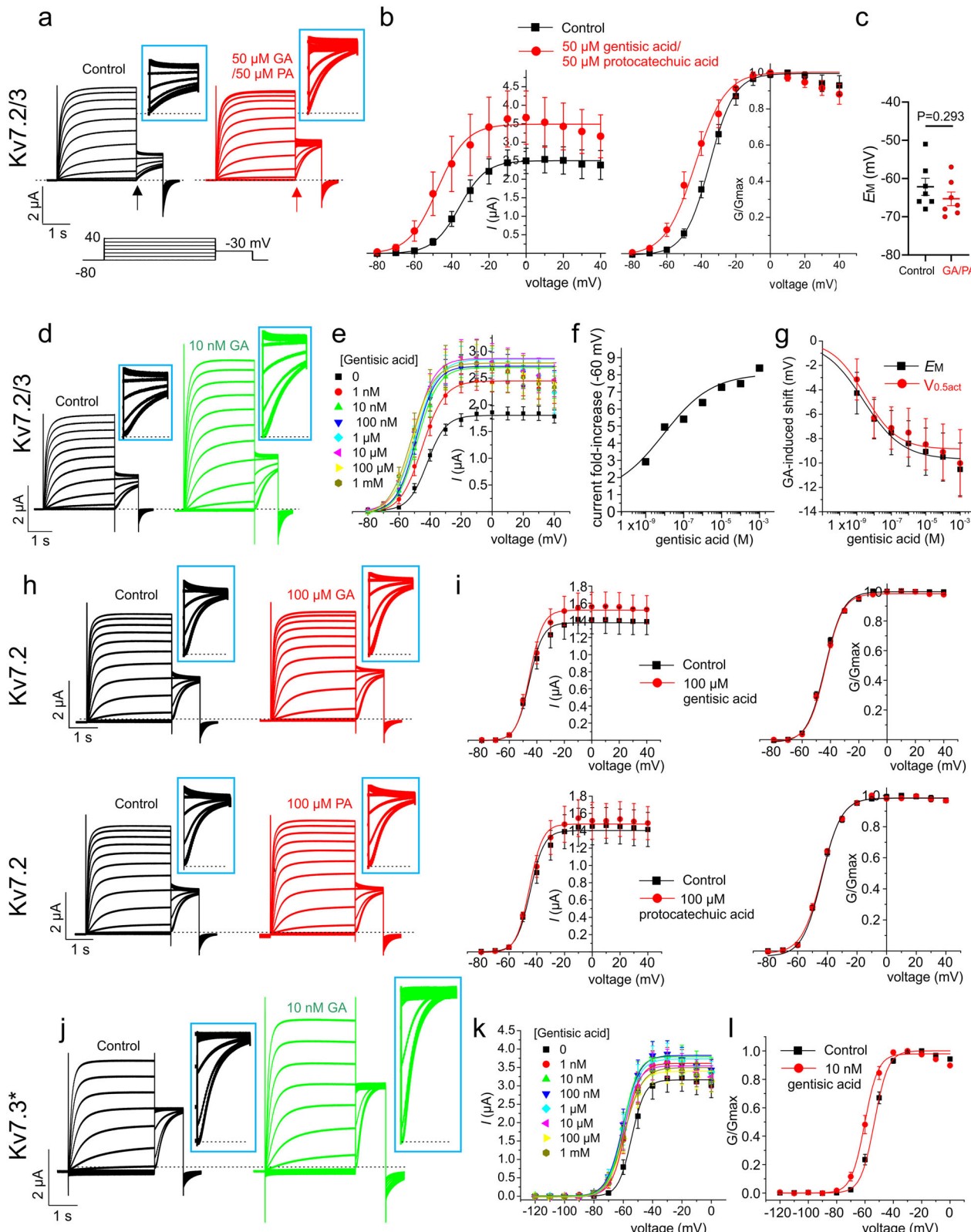

and the KCNE3 ancillary subunit (Fig. 5d–f), which in contrast are inhibited by tannic acid[29]. Further, we previously showed that Kv7.5 homomer function is not affected by GA (100 μM), which is, notably, also an active metabolite of aspirin[38,39]. Examining Kv channels outside the Kv7 family, we found that GA had no effects on prominent neuronal Kv channels Kv1.1, Kv1.2, and Kv2.1 (Fig. 5g–i).

**Gentisic acid ameliorates pentylenetetrazole-induced seizures in mice**

Because of the potent and selective Kv7.2/3 opening activity of GA, we tested its ability to ameliorate seizures induced by the chemoconvulsant pentylenetetrazole. Strikingly, we found that GA (2 mg kg$^{-1}$) more than doubled the latency to first seizure compared to mice treated with vehicle alone

**Fig. 4 | Gentisic acid is a potent Kv7.2/3 and Kv7.3 channel opener. a** Exemplar Kv7.2/3 traces recorded from oocytes in the absence (Control; black) or presence of 50 μM gentisic acid (GA) and 50 μM protocatechuic acid (PA) (red) using the voltage protocol indicated (lower inset). Scale bars center for the pair of traces; $n = 8$. **b** Mean peak tail current (left) and normalized tail current (G/Gmax) (right) for traces as in a (measured at arrows shown in a); $n = 8$, in the absence (black) or presence (red) of compounds indicated. **c** Mean unclamped oocyte membrane potential for Kv7.2/3-expressing oocytes as in (**a**); $n = 8$. **d** Exemplar Kv7.2/3 traces recorded from oocytes in the absence (Control; black) or presence of 10 nM gentisic acid (GA) (green) using the voltage protocol as in panel (**a**). Scale bars center for the pair of traces; $n = 8$ (1 nM, 1 mM), 10 (Control), 9 (all other gentisic acid doses). **e** Mean peak tail currents for Kv7.2/3 in the absence (black) or presence (red) of the various GA concentrations indicated, using protocol as in (**d**); $n$ values as in (**d**). **f** Mean fold-increase in peak tail current at −60 mV in response to the GA doses as in (**e**), recorded as in (**d**), fit with sigmoidal function; $n$ values as in (**d**). **g** Dose responses for mean shift in Kv7.2/3 $V_{0.5act}$ (red) and mean unclamped oocyte membrane potential for Kv7.2/3-expressing oocytes (black) recorded as in (**d**); $n$ values as in (**d**). **h** Exemplar Kv7.2 traces recorded from oocytes in the absence (Control; black) or

presence of gentisic acid (GA) (100 μM) (red) (upper) or presence of protocatechuic acid (PA) (100 μM) (red) (lower) using the voltage protocol as in panel (**a**). Scale bars lower left for the pair of traces; $n = 12$ (GA); 5 (PA) per group. **i** Mean peak tail current (left) and normalized tail current (G/Gmax) (right) for traces as in (**h**) (measured at arrows shown in a); $n = 12$ (GA); 5 (PA) per group, in the absence (black) or presence (red) of gentisic acid (GA) (100 μM) (upper) or protocatechuic acid (PA) (100 μM) (lower). **j** Exemplar Kv7.3* traces recorded from oocytes in the absence (Control; black) or presence of 10 nM gentisic acid (GA) (green) using the voltage protocol as in panel (**a**). Scale bars lower left for the pair of traces; $n = 7$. **k** Mean peak tail currents for Kv7.3* in the absence (black) or presence of the various GA concentrations indicated, using protocol as in (**a**); $n = 7$ except for the 1 mM group ($n = 6$). **l** Mean normalized tail current (G/Gmax) (right) for traces as in **j** (measured at arrows shown in a); $n = 7$, in the absence (black) or presence (red) of gentisic acid (GA) (10 nM). Error bars indicate SEM. $n$ indicates number of biologically independent oocytes. Statistical comparisons by $t$-test. Dashed lines indicate zero current level. Blue boxes contain tail currents for the corresponding traces expanded to preserve scale (test vs control).

($385 \pm 105$ s vs $152 \pm 14$ s, $n = 15$ vs 23, respectively, $p = 0.013$). However, increasing concentrations of GA had lesser effects on latency (10 mg kg⁻¹: $261 \pm 48$ s, $n = 23$; $p = 0.304$ vs vehicle; 20 mg kg⁻¹: $n = 8$; $156 \pm 25$ s, $p > 0.999$ vs vehicle), suggesting the possibility of counteracting effects at higher doses (Fig. 6).

### Gentisic acid requires Kv7.3-W265 for Kv7.2/3 opening

Previous studies showed that a negative electrostatic surface potential close to a carbonyl oxygen allows ligands such as retigabine to bind to S5 tryptophan W265 on Kv7.3 or its equivalent on Kv7.2 (W236) to open homomeric or heteromeric Kv7.2 and/or Kv7.3 channels[40–42]. We plotted the electrostatic surface potentials of all 12 portia tree compounds that we tested functionally and found that GA has the negative electrostatic surface potential best centered to its one carbonyl oxygen among all the compounds except for coumarin (Fig. 7a). We used unbiased docking to predict binding poses for GA on a model homomeric Kv7.3 structure (from Alphafold[43,44]) and found that GA is predicted to bind closely to the Kv7.3-W265 aromatic sidechain in a pi-stacking orientation (Fig. 7b, c).

Validating the docking prediction, substitution to leucine of Kv7.3-W265 eliminated effects of GA (100 μM) in Kv7.2/3 channels, as did substitution to leucine of both Kv7.3-W265 and Kv7.2-W236 (Fig. 7d, e). In contrast, Kv7.2/3 channels bearing only a Kv7.2-W236L substitution retained wild-type sensitivity to GA (Fig. 7d, e). Accordingly, GA hyperpolarized the $E_M$ of oocytes expressing Kv7.2/3 bearing only the Kv7.2-W236L substitution, but not the channels containing the Kv7.3-W265L substitution (Fig. 7f).

### GA structure-activity relationship

Structure-activity relationship (SAR) studies with GA, including aspirin and a range of other compounds each at 100 μM, indicated that both the carboxylic acid and the phenolic hydroxyls at positions 2 and 5 on the aromatic ring are important for maximal activity (Fig. 8a–g). Removal of the carboxylic acid gives inactive hydroquinone and the isomeric 2,3-dihydroxybenzoic acid is inactive at 100 μM. Compounds with fewer than two phenolic hydroxyls are inactive, including arbutin, aspirin, benzoic acid, 4-hydroxybenzoic acid and salicylic acid; in the case of arbutin, this recapitulated our prior finding[29]. Replacing benzoic acid with an acetic acid as in 3-hydroxyphenylacetic acid also abolished activity. Replacing the carboxylic acid with the neutral aldehyde or nitrile gave the inactive 2,5-dihydroxybenzaldehyde and 2,5-dihydroxybenzonitrile, respectively. Isosteric replacements for the carboxylic acid were tested including 3-hydroxyphenylboronic acid and 2,5-dihydroxyphenylsulfonic acid (calcium dobesilate). The boronic acid was devoid of activity; the lack of activity with the boronic acid (like 4-hydroxybenzoic acid) can be attributed to the lack of the phenolic hydroxyls at the 2 and 5-positions. However, one analog of GA did exhibit similar efficacy to that of GA—the sulfonic acid, calcium dobesilate (Fig. 8e, f, h).

## Discussion

*Thespesia populnea* (L.) Sol. ex Corrêa is a fast-growing tree (up to 12 m tall) found in pantropical coastal regions and has various popular names depending on the location, e.g., portia tree, seaside mahoe or *haiti-haiti* in the USVI, *miro* and *milo* in the Hawai'ian islands and Polynesia, Samoa, Tonga and American Samoa, and by many names, including *porasu*, in India. It is also referred to as Indian tulip tree, Pacific rosewood and Polynesian rosewood, among many other common names. Although there is still debate, it is typically considered to have originated in tropical Asia and then has spread (naturally and/or by humans) to other regions including the Caribbean and Africa; it may also be native to Indian Ocean and Pacific Island coasts, but its absence in the fossil record before early Polynesian human settlement suggests their role in its dispersal[45,46].

*T. populnea* is heavily used in indigenous medicine practices from various cultures; purported beneficial actions include wound-healing, antioxidant, anti-inflammatory, and bactericidal, and it has been used in traditional medicine for skin lesions and fractures[47]. Accordingly, controlled scientific studies have demonstrated anti-inflammatory actions of *T. populnea* bark, which also had analgesic activity in rodents; no acute oral toxicity was observed even at the highest dose (2 g kg⁻¹)[48]; similar effects were observed for leaf extract[49].

Another study showed anti-tumor and anti-inflammatory action in mice of *T. populnea* extract[50]. In support of its traditional use as a wound healing agent, in controlled scientific studies, *T. populnea* fruit was found to promote both excision and incision wound healing in rats[51], while petroleum ether and ethyl acetate fractions of *T. populnea* bark, in hydrogel formulations, were effective at wound healing, likely related to antioxidant and anti-inflammatory actions; compounds implicated in these activities were β-sitosterol, lupeol acetate, flavonoids, and anthocyanins including cyanidin and delphinidin, although no specific molecular correlates were firmly attributed[47]. However, gossypol, which is found in anti-inflammatory poultices made from *T. populnea* extract, has known anti-inflammatory properties, and has been suggested to contribute to this property in *T. populnea*[52].

Most importantly for the present study, *T. populnea* extract was also reportedly used to treat seizures, in the Rotuma group islands, which are 500 km north of the Fijian mainland[32,53]. In our previous studies of Kv7.2/3-opening plant extracts, we originally tested for this activity because of the plants' indigenous medicine use as anticonvulsants (e.g., *Coriandrum sativum* and *Mallotus oppositifolius*)[3,54]. In the present study, we identified *T. populnea* using an unbiased screen of 1444 extracts, and only subsequently learned of its traditional use by Polynesians as a folk anticonvulsant. We recently studied the action of tannic acid, including Kv7.2/3 opening as well as Kv1.3 inhibition, and examined its role in the analgesic and anti-inflammatory effects of medicinal plants such as witch hazel, which are applied topically, typically to abraded skin[30]. In the present study, we avoided selecting for plants in which tannic acid is the

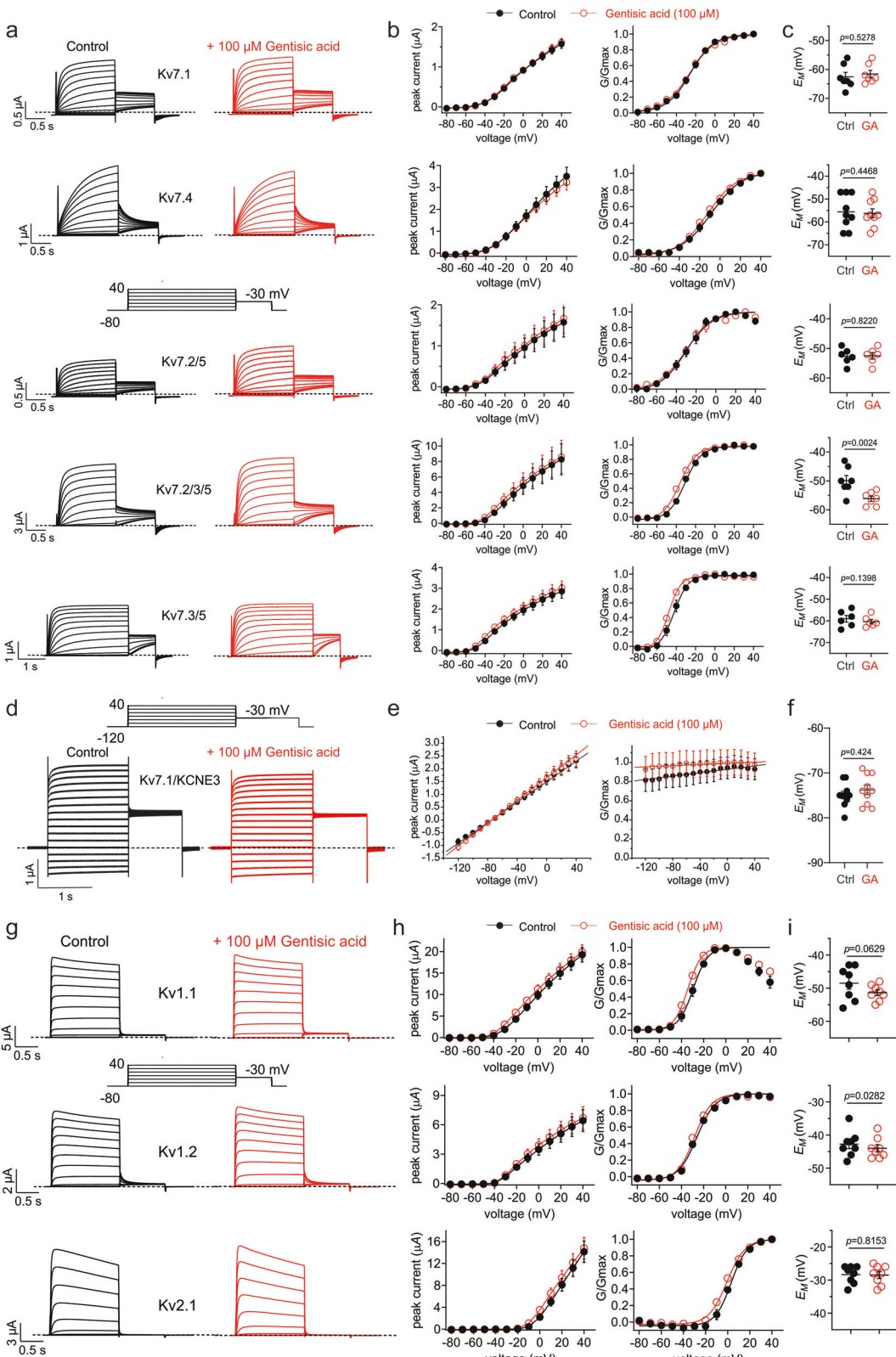

Kv7.2/3 opener, because of the limited usefulness those would have as an anticonvulsant, as explained earlier. By introducing this second filter, our screen was successful in identifying a traditional medicine anticonvulsant (*T. populnea*) and also a previously unknown action of gentisic acid as a Kv7.3 and Kv7.2/3 opener. Interestingly, GA is also a component of *Gentiana* species, such as *Gentiana oliverieri* Griseb. (Gentianaceae), a

flowering herb used as a folk anticonvulsant in Turkey and shown to be effective against maximal electroshock- and picrotoxin-induced seizures in mice[55]. GA has been studied for its potentially beneficial effects in relaxing guinea pig trachea (possibly involving BK channels, but for which the $EC_{50}$ was 20 μM, >1000-fold higher than the $EC_{50}$ for effects on Kv7.2/3) and reducing cardiac hypertrophy and preventing septic

**Fig. 5 | Gentisic acid is highly selective for Kv7.3 over the other Kv channels tested. a** Exemplar traces recorded from oocytes expressing the Kv channels indicated in the absence (Control; black) or presence of gentisic acid (100 μM) (red) using the voltage protocol indicated (inset). Scale bars lower left for each pair of traces; $n = 6$ (Kv7.2/5, Kv7.3/5), 7 (Kv7.1, Kv7.2/3/5), −9 (Kv7.4). **b** Mean peak prepulse current (left) and normalized tail current (G/Gmax) (right) for traces as in (**a**); $n$ values as in panel (**a**), in the absence (black) or presence (red) of gentisic acid (100 μM). **c** Mean unclamped oocyte membrane potential for oocytes as in (**a**); $n = 6$–9 per group. **d** Exemplar traces recorded from oocytes expressing Kv7.1/ KCNE3 in the absence (Control; black) or presence of gentisic acid (100 μM) (red) using the voltage protocol indicated (inset). Scale bars lower left for the pair of traces; $n = 10$ per group. **e** Mean peak prepulse current (left) and normalized tail current

(G/Gmax) (right) for traces as in (**d**); $n = 10$ per group, in the absence (black) or presence (red) of gentisic acid (100 μM). **f** Mean unclamped oocyte membrane potential for oocytes as in (**d**); $n = 10$ per group. **g** Exemplar traces recorded from oocytes expressing Kv1.1, Kv1.2 or Kv2.1 as indicated in the absence (Control; black) or presence (red) of gentisic acid (100 μM) using the voltage protocol indicated (inset). Scale bars lower left for the pair of traces; $n = 8$ per group. **h** Mean peak prepulse current (left) and normalized tail current (G/Gmax) (right) for traces as in (**g**); $n = 8$ per group, in the absence (black) or presence (red) of gentisic acid (100 μM). **i** Mean unclamped oocyte membrane potential for oocytes as in (**g**); $n = 8$ per group. Error bars indicate SEM. $n$ indicates number of biologically independent oocytes. Statistical comparisons by $t$-test. Dashed lines indicate zero current level.

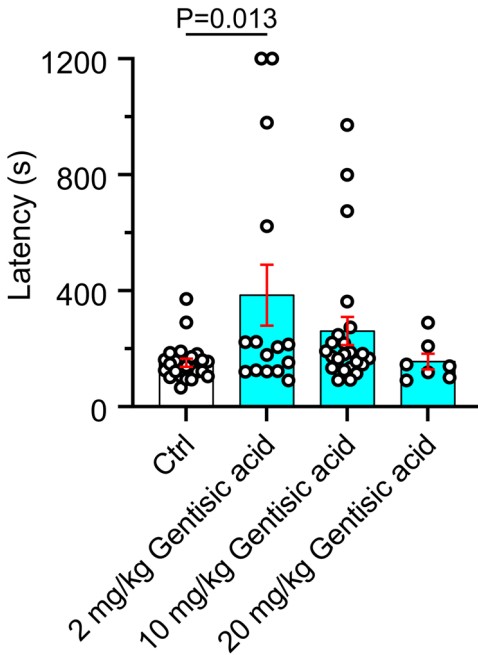

**Fig. 6 | Gentisic acid increases latency to first seizure in mice.** Error bars indicate SEM. Latency to first seizure was compared to the control (Ctrl) group using an ordinary one-way ANOVA with Dunnett correction for multiple comparisons. Graph shows mean ± SEM latency to first clonic seizure for mice injected IP with the agents indicated, 30 min prior to IP injection of pentylenetetrazole (80 mg/kg) to induce seizures. Ctrl = vehicle control (saline). $n = 23$ (Ctrl), 15 (2 mg kg$^{-1}$), 23 (10 mg kg$^{-1}$, 8 (20 mg kg$^{-1}$).

shock, but to our knowledge it has not been previously reported as an anticonvulsant or Kv7 channel modulator[56–58].

PA, which we found to be a less efficacious Kv7.2/3 opener than GA, was previously found to reduce neuronal death when given for several days post-seizure[59], but to our knowledge has not been evaluated as an anticonvulsant. In the GA SAR, we found that only one structurally related compound, the sulfonic acid calcium dobesilate, exhibited Kv7.2/3 opening activity similar to that of GA. The similar activity of the sulfonic acid compared to GA occurred despite the tetrahedral geometry of the sulfonate anion versus the planar carboxylate anion in GA, and despite the sulfonate anion being known to form weaker hydrogen bonds than the carboxylate anion[60]. Interestingly, calcium dobesilate is an FDA-approved vasoprotective drug used in the treatment of diabetic retinopathy and microvascular disorder, chronic venous insufficiency, varicose veins and hemorrhoids[61,62]. Calcium dobesilate is thought to act via a variety of mechanisms[63] but to our knowledge there are no previous reports of its direct interaction with ion channels. In future work we will further explore the ion channel activity profile of calcium dobesilate.

It is important to note that despite GA being a component of at least two different plants used as folk anticonvulsants, and its presence in many

other plant products that we eat, including apples, blackberries, kiwi, olives and avocados, much more research is needed before establishing its use in pure form as a safe and effective anticonvulsant. We found that its optimal efficacy was at 2 mg kg$^{-1}$ in mice, equivalent to 13 μM, while its efficacy waned at higher doses, possibly suggestive of a counteracting effect via a target with lower potency. In future work we will widen the seizure studies to different chemoconvulsant and genetic models, and a broader GA dose response, to more fully investigate the potential therapeutic window and possible clinical indications for GA. This would need to be further understood to optimize dosage and avoid potential proconvulsant effects, for example. Future studies can also include examination of other channel types and other proteins to provide additional information on specificity of the gentisic acid effects on Kv7.3. Another interesting and potentially safer therapeutic possibility is to use dietary modifications, increasing intake of GA-containing foods, but any potential strategies would need to be under the supervision of a qualified physician.

A recurring theme when studying isoform selectivity among Kv7 channels and beyond is that selectivity can be achieved despite the binding site being present on non-responding isoforms. Thus, Kv7.2, 4 and 5 each also possess the Kv7.3-W265 equivalent S5 tryptophan but are not responsive to GA even at concentrations >10,000 times the EC$_{50}$ for its effects on Kv7.3. Even with retigabine, responsiveness varies among Kv7.2-5 (and Kv7.1 is nonresponsive, because it lacks the S5 tryptophan). Studies that examine actual physical binding of openers to potassium channels are rare, although there are now highly illuminating high-resolution structures with some of the openers bound[64–67]. However, we previously showed that GABA, which we found to be a novel Kv7 opener, physically binds to Kv7.2-5 (but not Kv7.1) but only opens Kv7.3 and Kv7.5[68]. From this and other studies[69], we conclude that functional selectivity, as opposed to binding selectivity, is a major determinant of small molecule Kv7 opener selectivity, i.e., binding often occurs without this being transduced into a channel opening effect, depending on the isoform. Interestingly, here we found that GA exerts its maximal effects on Kv7.2/3 heteromers, inducing double the shift in V$_{0.5act}$ in Kv7.2/3 compared to homomeric Kv7.3, despite being inactive against Kv7.2. Further, we showed that one can eliminate Kv7.2-W236L in Kv7.2/3 heteromers and retain wild-type sensitivity to GA, while eliminating Kv7.2-W265L eliminates GA sensitivity in Kv7.2/3. We suggest that while Kv7.3 is the most sensitive in terms of GA potency (the EC$_{50}$ was 0.79 nM), Kv7.2/3 heteromers are more responsive in terms of transduction of the effects of binding into channel opening. We previously found that another plant metabolite, carnosic acid, is highly efficacious in opening Kv7.3 (-62 mV shift in V$_{0.5act}$ at 100 μM) albeit 1000-fold less potent than GA, but also does not open Kv7.2 and in contrast to GA, neither does carnosic acid open Kv7.2/7.3 channels[70]. While carnosic acid appears to reside in a lower binding site than GA (near the foot of the VSD), it is also predicted to bind to residues in Kv7.3 that are conserved in Kv7.2[70]. One working hypothesis is that a higher sensitivity in either the gating apparatus or voltage sensing apparatus in Kv7.3 endows it with greater responsivity than Kv7.2 to binding of molecules such as GA and carnosic acid.

In conclusion, unbiased screening for Kv7.2/3 openers with a filter to eliminate the, in this case, unwanted compound tannic acid has selected a Polynesian folk medicine botanical anticonvulsant, in turn leading to

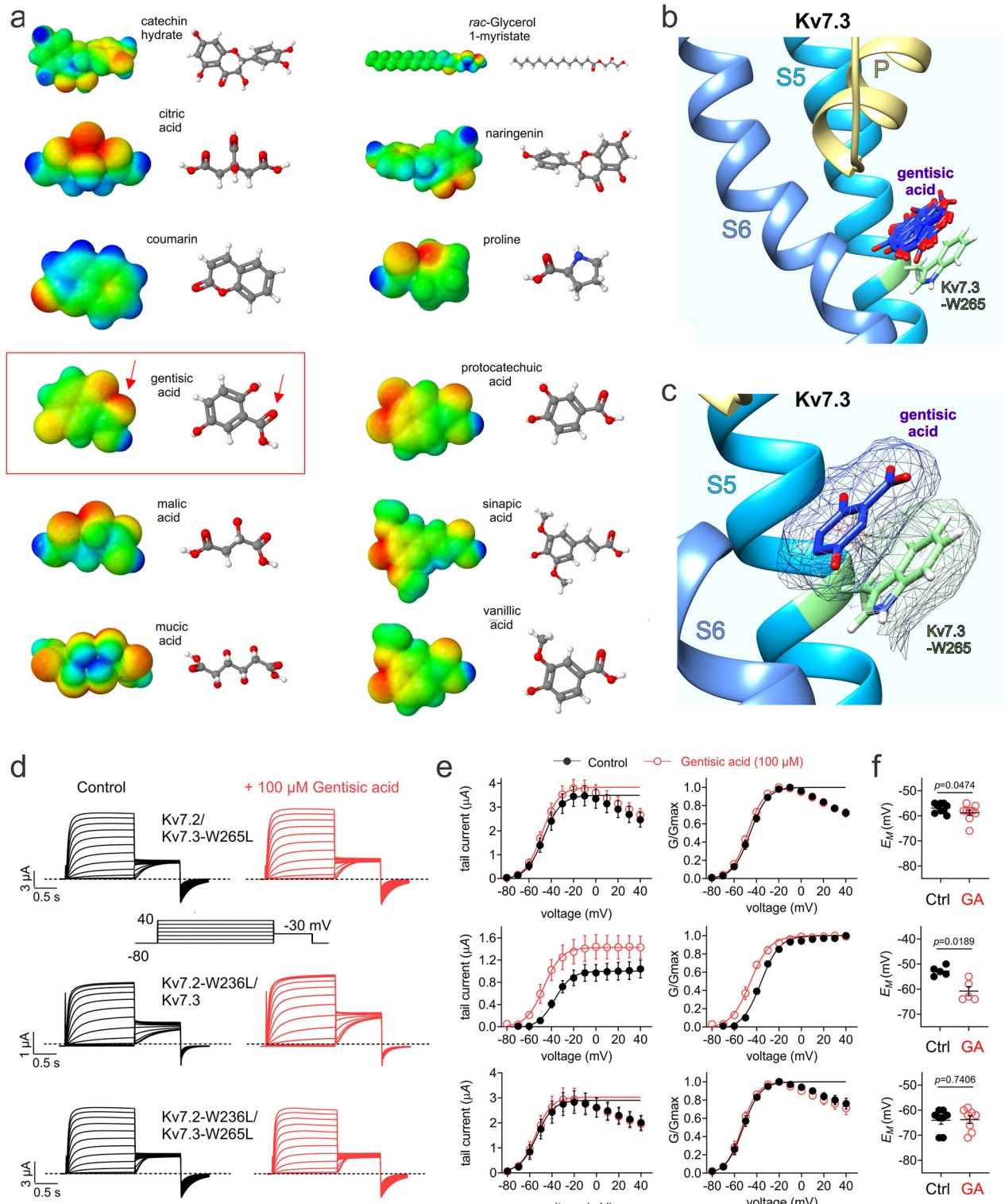

**Fig. 7 | Kv7.3-W265 is required for gentisic acid opening of Kv7.2/3.**
**a** Electrostatic surface potential (left; red = negative, blue = positive) and structure (right; red = oxygen, blue = nitrogen) for portia tree compounds tested herein.
**b** Unbiased SwissDock predicted binding of gentisic acid (blue; multiple binding poses indicated) to Kv7.3-W265 (seafoam green). Kv7.3 transmembrane segments (Sx) and pore (P) labeled. **c** Close-up of one gentisic binding pose from panel (**b**), with wireframe surface plots shown. **d** Mean traces for Kv7.2/3 mutant combinations as indicated, expressed in oocytes in the absence (Control) or presence of gentisic

acid (100 µM). Scale bars lower left; voltage protocol center inset; $n = 9$ (Kv7.2/Kv7.3-W265L), 5 (Kv7.2-W236L/Kv7.2), 8 (Kv7.2L-W236L/Kv7.3-W265L).
**e** Mean raw tail current/voltage relationship (left) and normalized tail current (G/Gmax)/voltage relationship (right) for traces as in (**d**); $n$ values as in (**d**). **f** Mean unclamped oocyte membrane potential for mutant Kv7.2/3-expressing oocytes as in (**d**); $n$ values as in (**d**). Error bars indicate SEM. $n$ indicates number of biologically independent oocytes. Statistical comparisons by $t$-test.

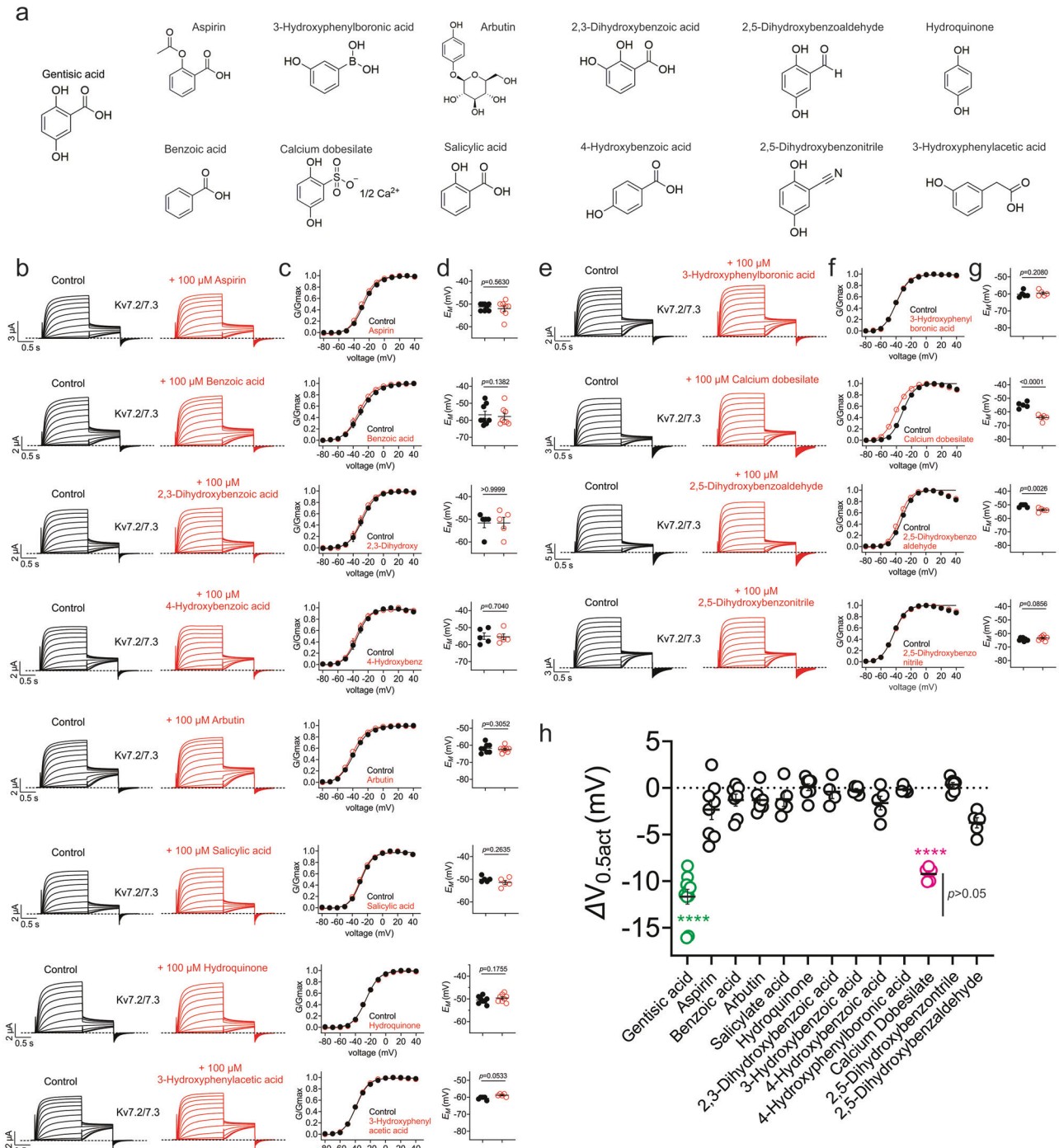

**Fig. 8 | The gentisic acid SAR for Kv7.2/3 channel opening. a** Structures of compounds in the GA SAR. **b** Mean traces recorded from oocytes expressing Kv7.2/3 in the absence (Control; black) or presence of GA analogs (100 μM) (red) using the voltage protocol from Fig. 5a. Scale bars lower left for each pair of traces; $n = 5$ (2,3-Dihydroxybenzoic acid, 4-Hydroxybenzoic acid, salicyclic acid, 3-Hydroxyphenylacetic acid) or 8 (arbutin, aspirin, benzoic acid, hydroquinone) per group. **c** Mean peak normalized tail current (G/Gmax) for traces as in (**b**); $n$ values as in (**b**), in the absence (black) or presence (red) of compounds indicated (100 μM). **d** Mean unclamped oocyte membrane potential for oocytes as in (**b**); $n$ values as in (**b**). **e** Mean traces recorded from oocytes expressing Kv7.2/3 in the absence (Control; black) or presence of GA analogs (100 μM) (red) using the voltage protocol from Fig. 5a. Scale bars lower left for each pair of traces; $n = 5$ (3-Hydroxyphenylboronic acid, calcium dobesilate,

2,5-Dihydroxybenzoaldehyde) or 8 (2,5-Dihydroxybenzonitrile) per group. **f** Mean peak normalized tail current (G/Gmax) for traces as in (**e**); $n$ values as in (**e**), in the absence (black) or presence (red) of compounds indicated (100 μM). **g** Mean unclamped oocyte membrane potential for oocytes as in (**e**); $n$ values as in (**e**). **h** Mean Kv7.2/3 $\Delta V_{0.5act}$ for the compounds tested in panels (**b–g**) ($n$ values as in **b** and **e**). ****$P < 0.0001$ (green) GA vs all other compounds except calcium dobesilate; ****$P < 0.0001$ (magenta) calcium dobesilate vs all other compounds except GA; GA vs calcium dobesilate: $P > 0.05$. GA data analyzed from traces as in Fig. 3a ($n = 10$). Error bars indicate SEM. $n$ indicates number of biologically independent oocytes. Statistical comparisons by $t$-test except panel (**h**), which was by ANOVA with multiple comparisons. Dashed lines indicate zero current level.

discovery of a highly potent and Kv7.3-selective compound (GA) with anticonvulsant properties. Our findings provide a molecular rationale for the use of *T. populnea* extract by Rotuma group islanders as an anticonvulsant. Importantly, our data underscore the ingenuity of indigenous people of the South Pacific in utilizing the resources in their natural environment, and it is quite possible that they or their ancestors brought *T. populnea* seeds from Asia to their new world for a specific purpose or purposes, perhaps even as medicine[71,72].

## Methods

All chemicals central to this study were purchased from Sigma Aldrich (St. Louis, MO, US) and were of ACS reagent grade purity or higher. Species identification of plants central to this study was certified "Research Grade" at iNaturalist.com.

### Collection of plant samples

We collected, between 2019 and 2022, aerial parts of plants under permit from Mojave National Preserve (study # MOJA-00321), Yosemite National Park (study # YOSE-00839), Santa Monica Mountains National Recreation Area (study # SAMO-00192), Muir Woods National Monument (study # MUWO-00035), Santa Cruz Island and Santa Rosa Island (study # CHIS-0023), and Boyd Deep Canyon (Indian Wells, CA) (permit through Philip L. Boyd Deep Canyon Desert Research Center, University of California, Riverside, CA) in California, US. In USVI, we collected from Virgin Islands National Park, St. John (study # VIIS-20001), Salt River Bay National Historical Park and Ecological Preserve, St. Croix (study # SARI-00056) and Buck Island Reef National Monument, St. Croix (study # BUIS-00103). Plant samples were collected in a manner designed to not kill the remaining plant, sealed in *Ziploc* bags (SC Johnson, Racine, WI, US), kept cold, and frozen as soon as possible. Other plant extracts were made from plants purchased from Crimson Sage Nursery (Orleans, CA) and grown in the corresponding author's garden in Irvine CA, from Mountain Rose Herbs (Eugene, OR) or from Mother's Market (Irvine, CA). Plant samples were stored in −20 °C freezers until extraction.

### Preparation of plant extracts

Leaves and flowers were pulverized using a bead mill with porcelain beads in batches in 50 ml tubes (Omni International, Kennesaw, GA, United States), then the homogenates resuspended in 80% methanol/20% water (100 ml per 5 g solid) and incubated for 48 h at room temperature, with occasional inversion to resuspend the particulate matter. We then filtered the extracts through Whatman filter paper #1 (Whatman, Maidstone, UK), removed the methanol using evaporation in a fume hood for 24–48 h at room temperature, centrifuged extracts for 10 min at 15 °C, 4000 RCF to remove the remaining particulate matter, followed by storage (−20 °C). On the day of electrophysiological recording, we thawed the extracts and diluted them 1:50 in bath solution (see below), equivalent to 5 mg fresh plant matter starting material/ml, immediately before use.

### High-throughput screening for Kv7.2/3 activation

Plant extracts were applied to human Kv7.2/3 channels expressed in HEK293 cells using a FLIPR potassium assay kit and a Fluorescence Imaging Plate Reader (FLIPR$^{TETRA^-}$) instrument. All chemicals used in this project were purchased from Sigma–Aldrich (St. Louis, MO) unless otherwise noted and were of ACS reagent grade purity or higher. Stock solutions of the positive controls were prepared in dimethyl sulfoxide (DMSO) or deionized water, aliquoted, and stored frozen. Plant extracts were prepared in buffer and frozen until dilution into the assay buffers on test day. The ability of each plant extract to act as a Kv7.2/3 channel opener was evaluated in a thallium (potassium ion surrogate, Molecular Devices) flux assay by Charles River Laboratories (Cleveland, OH, US). The assay was performed with the FLIPR potassium assay kit (Molecular Devices) according to the manufacturer's instructions. For dye loading, the growth media was removed and replaced with 20 μl of dye loading buffer for 60 min at room temperature. For stimulation (agonist mode):

5× (5 μL) plant extract, vehicle, or control article solutions prepared in the stimulation buffer (K$^+$-free buffer with 5 mM Tl$^+$) was added to each well for ~5 min. The agonist effects of the plant extract or control articles on KCNQ2/3 channels were evaluated. The positive control was Flupirtine (9 concentrations). Data acquisition was performed via the ScreenWorks FLIPR control software that is supplied with the FLIPR System (MDS-AT). Data were analyzed using Microsoft Excel 2013 (Microsoft Corp., Redmond, WA). For each well, the raw kinetic data were reduced to the maximum or Area Under Curve fluorescence after subtracting bias and possibly applying the negative control correction. Reduced data were analyzed as follows:

For each assay plate, a Z' factor and Signal Window (SW) were calculated:

$$Z' \text{ factor} = ((\text{[Agonist Control mean]} - 3 \times \text{[Agonist Control STDEV]}) - (\text{[Vehicle Control mean]} + 3 \times \text{[Vehicle Control STDEV]})) / (\text{[Agonist Control mean]} - \text{[Vehicle Control mean]})$$

$$SW = ((\text{[Agonist Control mean]} - 3 \times \text{[Agonist Control STDEV]}) - (\text{[Vehicle Control mean]} + 3 \times \text{[Vehicle Control STDEV]})) / \text{[Agonist Control STDEV]}$$

Where the stimulation buffer was dispensed to Vehicle Control wells and a high concentration of agonist positive control was dispensed to Agonist Control wells.

Concentration-response curves were fitted to the agonist positive control.

Reduced data from test article wells were normalized to the vehicle and agonist control means on each plate and expressed as normalized percent activation:

$$\text{Normalized\%Activation} = (\text{[individual well RLU]} - \text{[Vehicle Control mean]}) / (\text{[Agonist Control mean]} - \text{[Vehicle Control mean]})$$

where individual well RLU = the relative light units for each well to which test article is dispensed. A significance threshold of 3 standard deviations from the vehicle control mean was calculated:

$$\text{Significance Threshold} = 3x\text{[Vehicle Control STDEV]} / (\text{[Agonist Control mean]} - \text{[Vehicle Control mean]})$$

Concentration-response curves for positive agonist controls for each plate were also conducted. The positive control results confirmed the sensitivity of the test systems to agonists. The test and control samples were prepared in the stimulation buffer (a combination of low Cl$^-$ buffer, 5 mM Tl$_2$SO$_4$ and water). The signal elicited in the presence of the positive agonist control (30 or 100 μM Flupirtine) was set to 100% activation and the signal from the vehicle (stimulation buffer) was set to 0% activation.

### High-throughput screening for Kv1.3 inhibition

Chemicals used in solution preparation were purchased from Sigma–Aldrich unless otherwise noted and were of ACS reagent grade purity or higher. Stock solutions of plant extracts and the positive controls were prepared in water and stored frozen, unless otherwise specified. Reference compound concentrations were prepared fresh daily by diluting stock solutions into a HEPES-buffered physiological saline (HB-PS) (composition in mM): NaCl, 137; KCl, 4.0; CaCl2, 4.8; MgCl2, 1; HEPES, 10; Glucose, 10; pH adjusted to 7.4 with NaOH. To minimize run-down of the Kv1.3 channel currents 0.3% DMSO was added in all reference, plant extract and control solutions. The plant extracts (diluted to 2% and 0.2%, equivalent to 5 and 0.5 mg fresh plant matter starting material/ml, respectively, were loaded into 384-well polypropylene compound plates and placed in the plate well of an automated patch-clamp (APC) system, SyncroPatchTM 384PE (SP384PE; Nanion Technologies, Livingston, NJ) immediately before application to Chinese Hamster Ovary (CHO) cells (strain source, ATCC Manassas, VA; sub-strain source, ChanTest Corporation, Cleveland, OH, US) expressing human Kv1.3. Screening was conducted by Charles River Laboratories.

Extracellular buffer was loaded into the wells of the Nanion 384-well Patch Clamp (NPC-384) chips (60 µl per well). Then, cell suspension was pipetted into the wells (20 µL per well) of the NPC-384 chip. After establishment of a whole-cell patch-clamp configuration, membrane currents were recorded using the patch clamp amplifier in the SP384PE system. Plant extracts were applied to naïve cells ($n = 3$, where $n =$ the number cells/concentration). Each application consisted of addition of 40 µl of 2× concentrated test article solution to the total 80 µl of final volume of the extracellular well of the NPC-384 chip. Duration of exposure to each test article concentration was five (5) minutes. The intracellular solution was (in mM): KCl, 70; KF, 70; MgCl2, 2; EGTA, 2.5; HEPES, 10; pH adjusted to 7.2 with KOH. In preparation for a recording session, the intracellular solution was loaded into the intracellular compartment of the NPC-384 chip. The extracellular solution was the HB-PS solution described above.

Kv1.3 channel currents were elicited using test pulses with fixed amplitudes: depolarization pulse to +20 mV amplitude, 200 ms duration from the holding potential of –90 mV. The test pulses were repeated with frequency 0.1 Hz: 3 min before (baseline) and 5 min after test articles addition. Kv1.3 channel current amplitudes were measured at the peak and at the end of the step to +20 mV. The positive control antagonist used was 4-aminopyridine, prepared as a 1 M stock in water; test concentrations were 1, 3, 10, 30, 100, 300, 1000 and 3000 µM.

### Channel subunit cRNA preparation and *Xenopus laevis* oocyte injection for manual two-electrode voltage-clamp (TEVC) electrophysiology

We generated cRNA transcripts encoding human Kv1.1 (KCNA1), Kv1.2 (KCNA2), Kv2.1 (KCNB1), Kv7.1, Kv7.2, Kv7.3, Kv7.4, Kv7.5 (KCNQ1-5), and KCNE3 (MiRP2) by in vitro transcription using the mMessage mMachine kit (Thermo Fisher Scientific, Waltham, MA, USA) according to manufacturer's instructions, after vector linearization, from cDNA subcloned into expression vectors (pTLNx, pXOOM and pMAX) incorporating *Xenopus laevis* β-globin 5' and 3' UTRs flanking the coding region to enhance translation and cRNA stability. We injected defolliculated stage V and VI *Xenopus laevis* oocytes (Xenoocyte, Dexter, MI, USA) with the channel cRNAs (0.3–10 ng) and incubated the oocytes at 16 °C in ND96 oocyte storage solution containing penicillin and streptomycin, with daily washing, for 1–4 days prior to two-electrode voltage-clamp (TEVC) recording. Mutant channel cDNAs were generated by GenScript Biotech (Piscataway, NJ).

### Two-electrode voltage clamp (TEVC)

We conducted TEVC at room temperature with an OC-725C amplifier (Warner Instruments, Hamden, CT, USA) and pClamp10 software (Molecular Devices, Sunnyvale, CA, USA) 1–4 days after cRNA injection. We visualized oocytes in a small-volume oocyte bath (Warner) using a dissection microscope for cellular electrophysiology. We studied the effects of plant extracts and constituents solubilized directly in bath solution (in mM): 96 NaCl, 4 KCl, 1 MgCl$_2$, 1 CaCl$_2$, 10 HEPES (pH 7.6). We introduced extracts or compounds into the oocyte recording bath by gravity perfusion at a constant flow of 1 ml per minute for 3 min prior to recording. Pipettes (1–2 MΩ resistance) were filled with 3 M KCl. We recorded currents in response to voltage pulses between −120 mV or −80 mV and +40 mV at 10 mV intervals from a holding potential of −80 mV, to yield current-voltage relationships. We analyzed data using Clampfit (Molecular Devices) and Graphpad Prism software (GraphPad, San Diego, CA, USA), stating values as mean ± SEM. We calculated the voltage dependence of activation ($V_{0.5}$) by measuring currents at a voltage pulse of −30 mV (Kv7) or −50 mV (Kv1) immediately following prepulse voltages between −120 mV or −80 mV and +40 mV. We plotted raw or normalized tail currents versus prepulse voltage and fitted them with a single Boltzmann function:

$$g = \frac{(A_1 - A_2)}{\{1 + \exp[V_{\frac{1}{2}} - V/Vs]\}y + A_2} \tag{1}$$

where $g$ is the normalized tail conductance, $A_1$ is the initial value at $-\infty$, $A_2$ is the final value at $+\infty$, $V_{1/2}$ is the half-maximal voltage of activation and $V_s$ the slope factor.

### Seizure studies

The mouse study was performed under an approved Institutional Animal Care and Use Committee protocol at the University of California, Irvine. Male C57Bl/6 mice at 2 months of age were injected intraperitoneally (IP) with either vehicle (saline) or gentisic acid at 2, 10 or 20 mg/kg (pH7.5). After 30 min, mice were next injected IP with pentylenetetrazole (80 mg/kg) and then observed by a scorer blinded to the treatment, who measured the latency to first clonic seizure.

### In silico docking

We plotted and viewed chemical structures and electrostatic surface potential using Jmol, an open-source Java viewer for chemical structures in 3D: http://jmol.org/. For in silico ligand docking predictions of binding to Kv7.3, we performed unguided docking to predict potential binding sites, using SwissDock with CHARMM forcefields[73,74] and the AlphaFold-predicted[43,44] human Kv7.3 monomer structure. We prepared channel structures for docking using DockPrep in UCSF Chimera (https://www.rbvi.ucsf.edu/chimera)[75], with which we also generated docking figures.

### Statistics and reproducibility

All values are expressed as mean ± SEM. One-way ANOVA (with Dunnett correction for multiple comparisons in the case of seizure studies) or *t*-test was applied for all tests; all *p* values were two-sided. Electrophysiological data were confirmed in at least two batches of oocytes. Biological replicates are defined as numbers of oocytes; sample sizes are given in the figure legends.

### Reporting summary

Further information on research design is available in the Nature Portfolio Reporting Summary linked to this article.

## Data availability

Source data for Figs. 1–8 are available at Dryad data repository: https://doi.org/10.5061/dryad.547d7wmhn.

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

## Acknowledgements
This study was supported by the National Institutes of Health, National Institute of General Medical Sciences (GM130377) and a Susan Samueli Integrative Health Institute, Samueli Scholarship to GWA. This work was performed (in part) at the University of California Natural Reserve System, Philip L. Boyd Deep Canyon Desert Research Center, Reserve https://doi.org/10.21973/N3V66D. We thank Grey Arena, Emma Laskey, Alexandra Kookootsedes and Elliot Gunnison for logistical support, collection, and identification of plants in Muir Woods, Esha Kaur, Kevin Tran, Adam Buie, Kristina Mai, Catherine Tran, Deanna Tran, Henry Wu, Samy Haidar, and William Deacon (University of California, Irvine) for performing methanolic plant extractions. We sincerely thank the iNaturalist community for their plant identification suggestions, Kaitlyn E. Redford (University of California, Irvine) for helping with plant collection in Yosemite National Park, Dr. Ryan Yoshimura for organization of the plant extracts and the associated database, Dr. Derk Hogenkamp for chemical structure plots and an explanation of the SAR, and Dr. Tallie Baram for fruitful discussions about Gentian.

## Author contributions
G.W.A. conceived the study, collected and coordinated identification of the plants, conducted and analyzed the majority of experiments, prepared figures, wrote the manuscript and obtained project funding; R.W.M. helped with plant collection, conducted TEVC experiments and data analysis, prepared solutions for blinded seizure experiments, prepared figures and edited the manuscript.

## Competing interests
The authors declare no competing interests.
