## [Transparent Peer Review file · Communications Chemistry]

Discovery of a potent, Kv7.3-selective potassium channel opener from a Polynesian traditional botanical anticonvulsant

Corresponding Author: Professor Geoffrey Abbott

Version 0:

Reviewer comments:

Reviewer #1

(Remarks to the Author)

This manuscript describes the discovery of a potent Kv7.3-selective potassium channel openers from a polynesian traditional botanical anticonvulsant. The primary active component appears to be gentisic acid which is an extremely potent activator and very selective at least when compared to other KV7 channels and a small number of other KV channel types, effective at nM concentrations. As such, the bulk of the manuscript focuses on the actions of gentisic acid in isolation.

The content of the manuscript is fairly straightforward but it is clear from the raw data traces that the quality of the data obtained is high and robust. It is the potency and selectivity of gentisic acid at 'activating' KV7.3 channels that marks out this work as novel and of some considerable interest and prove of value to researchers working both directly in the field and in the wider context of epilepsy research. I have some points, below, which the authors should consider.

- 1) The introduction is dominated by references to the groups own previous work. So, for example, the 1st 10 references cited are all from this group. Perhaps the authors should consider a wider range of references to introduce the manuscript.
- 2) Lines 89-90: There is a need for a reference to the use of the plant used as an anticonvulsant in polynesian traditional medicine and more detail here.
- 3) Some justification is required as to the focus on KV 7.2/7.3 and the use of KV 1.3 as a control. Why choose KV 1.3 particularly the authors need to justify the lack of screening against other potassium channels when talking about selected activation. There is a short description of other KV channels tested on lines 195-196 but this should be made more prominent and perhaps extend to a wider range of other ion channels.
- 4) Line 105 - justify the exclusion criteria described here
- 5) Line 115-116 - this seems rather vague account here.
- 6) The legend for Figure 1 refers to a previous figure in reference 25. However reference 25 has not been published anywhere so it is impossible to compare the earlier version of the figure with the one shown here.
- 7) Lines 127-134 are really a description of the methods (or rather the justification for the methods chosen) for quantification and, as such, should probably sit in the method section
- 8) Figure 2 could be rearranged to show the most important effect first namely the effect of Portia tree extra as this is a subject or the focus of the rest of the paper - this might be more effective than putting a box around these data. This is, after all, the key finding of the paper.
- 9) There are some anomalies in the data in figure 3. The control resting membrane potential varies quite considerably from one set of experiments to another so that although gentisic acid hyperpolarises the membrane potential by about 10 millivolts this really only takes it to the resting value seen in the experiments with coumarin for example. One could argue that in these coumarin experiments the membrane potential is already hyperpolarized before the addition of the drug and therefore the compound would not be able to hyperpolarize further. Despite this, the effect of gentisic seems clear enough to warrant further study as described subsequently in the paper.
- 10) I don't think the authors make enough of the potency of gentisic acid. In many experiments 10-100 uM concentrations are used, yet figure 4 d,e show that 10 nM is almost maximally effective. Comparing 10 nM gentisic acid on KV7.3 with 10s of

uM on other channels would be extremely persuasive.

11) The raw data traces show examples of beautiful current recordings for which the authors should be commended. However, the effects of compounds are measured/quantified on steps back to -30 mV following the test steps which are only a small fraction of the traces shown. These steps to -30 mV bring out the effects of the compounds most clearly but are perhaps lost in the raw data traces. Could this section of the traces be magnified with, perhaps, the whole trace shown as insets to bring out the effects of compounds?

12) There is a bell shaped effect of gentisic acid seen on seizures. Why do the authors think this is so? How do the concentrations used here in mg/kg relate to the concentrations used in the electrophysiological experiments. The latter suggests that really low concentrations (10 nM) have a large effect and that whilst there is a saturation of the effect on KV7.3 containing channels there is not a reversal at high concentrations.

13) Lines 305-324. Perhaps the authors could speculate further as to why the compound is so selective for KV7.3 containing channels and why KV7.2/7.3 channels are more responsive in terms of transducing the effect of binding into channel opening.

Reviewer #2

(Remarks to the Author)

The paper by Abbott and Manville describes the identification of gentisic acid (GA) as a novel and potent Kv7.2/3 channel activator present in the flower/leaf extract of portia tree (*Thespesia populnea*). GA was identified following an extensive high throughput screen (thallium flux assay) of 1444 plant extracts and follow up validation testing on Kv7.2/3 channels expressed in *Xenopus* oocytes. Protocatechuic acid (PA) was also identified yet had lesser effects compared to GA. The screening protocol importantly excluded plant extracts with known high tannic acid levels, which itself has Kv7.2/3 channel activity that would otherwise occlude the ability to identify novel compounds.

The search for potent and selective Kv7.2/3 channel openers are of clinical interest for use as anticonvulsant agents in the treatment of epilepsies and seizure disorders. GA demonstrated high potency, with EC50's of 2-15 nM for different channel effects, and also demonstrated high selectivity with little to no effect on other Kv channels tested. Mechanistically, evidence is provided the GA binds to W265 in the Kv7.3 S5 segment via pi-stacking, where the W265L mutation resulted in a complete loss of GA-induced channel activity. This site has been previously implicated in the binding and actions of retigabine.

The experimental findings of GA actions on Kv7.2/3 channels is compelling and will be of general widespread interest. The experimental findings of GA actions as an anticonvulsant in the mouse seizure model, however, were less convincing. Pre-administration of GA (i.p. injection) was found to delay the acute onset of seizures induced by pentylenetetrazole (PTZ) injected 30 minutes later in mice. This effect, however, was only statistically produced at one dose tested (2 mg/kg), where higher doses (10 or 20 mg/kg) did not produce the same effect. From the data presented in figure 6, most mice did not respond at the 2 mg/kg dosage, where a small subgroup of 'responders' demonstrated a latency greater than the control latency value (~150 s). Thus, the acute anticonvulsant activity of GA that is presented, is currently the weakest part of the paper where the claim GA "ameliorates pentylenetetrazole-induced seizures in mice" is not overly convincing. Additional experiments or further discussion with possible explanations would help. Nevertheless, a more comprehensive testing of GA antiseizure activity (acute and chronic) is undoubtedly anticipated and expected to be the subject of follow up pre-clinical studies. The discussion of the portia tree being used as a Polynesian traditional anticonvulsant is quite interesting given the findings of GA and PA on Kv7.2/3 channels.

Minor

Delete line 848 in the figure legend for figure 6.

Reviewer #3

(Remarks to the Author)

This report extends the work of this group in identifying new modulators of potassium channel action in extract from plants collected in California and the US Virgin Islands. In this report, their focus is on activators of the voltage-gated potassium Kv7.2/3, which is a target for anticonvulsant drug development. They included a counter-screen against Kv1.3 to select against inhibitors of this channel. Seven extracts were identified that increased Kv7.2/3 thallium flux by 76-150%, with Kv1.3 inhibition <93%.

As a portia tree extract showed a promising profile, the effects of 12 of the most abundant compounds present in this extract, as identified by others, were investigated. Gentisic acid (GA) showed an EC50 for increasing Kv7.2/3 current at -60 mV of 15.6 nM, and EC50 values for shifting V0.5act and EM of Kv7.2/3 of 2.8 and 2.4 nM, respectively. GA did not affect Kv7.2 homomers but did open Kv7.3* channels.

It was then shown that GA ameliorated pentylenetetrazole-induced seizures in mice. GA at 2 mg/kg more than doubled the latency to first seizure compared to mice treated with vehicle alone, although increasing concentrations of GA had weaker effects on latency. As acknowledged in the Discussion, this would need to be thoroughly understood to optimize dosage and avoid potential proconvulsant effects before GA could be considered for use as an anticonvulsant.

Others have proposed that retigabine interacts with S5 W265 on Kv7.3 or its equivalent on Kv7.2 (W236). Docking

conducted in this study suggests that GA could also interact with Kv7.3-W265. Mutation of Kv7.3-W265 to Leu eliminated the effects of GA (100 μ M) on Kv7.2/3 channels, as did mutation to Leu of both Kv7.3-W265 and Kv7.2-W236. It is remarkable that such limited interactions between GA and the channel could confer selectivity. As indicated in the Discussion, there seems to be something of a disconnect between binding selectivity and functional selectivity in determining small molecule Kv7 opener selectivity, although the molecular basis for this is not well understood.

While this study is certainly of interest, I question whether it is appropriate for a chemistry journal, as opposed to one focusing on, for example, pharmacognosy or molecular pharmacology. There is essentially no chemistry in the current manuscript.

Line 246-247: 'T. populnea wood is used to construct furniture, musical instruments, kitchenware 247 and canoe paddles, etc., while the bark makes effective rope and caulk.' This is irrelevant and should be deleted. The same applies to lines 334-340 in the Discussion.

Data Availability: Source data for Figures 1-7 and Supplementary Figure 1 are available at Dryad data repository: xxxxxxxx. The repository information is required.

Version 2:

Reviewer comments:

Reviewer #1

(Remarks to the Author)

. I have looked at the revised manuscript and the comments of the authors in response to my review of their initial submission and I am happy that the manuscript is now suitable for publication

The authors sincerely thank the reviewers for their thorough and highly positive reviews. Please see our point-by-point responses below. **All page and paragraph numbers described in my responses refer to the marked up (changes tracked) version of the manuscript.**

Reviewer #1 (Remarks to the Author):

This manuscript describes the discovery of a potent Kv7.3-selective potassium channel opener from a polynesian traditional botanical anticonvulsant. The primary active component appears to be gentisic acid which is an extremely potent activator and very selective at least when compared to other KV7 channels and a small number of other KV channel types, effective at nM concentrations. As such, the bulk of the manuscript focuses on the actions of gentisic acid in isolation.

The content of the manuscript is fairly straightforward but it is clear from the raw data traces that the quality of the data obtained is high and robust. It is the potency and selectivity of gentisic acid at 'activating' KV7.3 channels that marks out this work as novel and of some considerable interest and prove of value to researchers working both directly in the field and in the wider context of epilepsy research. I have some points, below, which the authors should consider.

We thank Reviewer 1 for this highly positive assessment.

1) The introduction is dominated by references to the groups own previous work. So, for example, the 1st 10 references cited are all from this group. Perhaps the authors should consider a wider range of references to introduce the manuscript.

We have changed the references as requested.

2) Lines 89-90: There is a need for a reference to the use of the plant used as an anticonvulsant in polynesian traditional medicine and more detail here.

We have added a reference (which is also in the Discussion) - there are no more details available as Rotuman medicine has not been extensively studied; certainly a subject for future ethnobotanical investigation into Rotuman folk medicine practices.

3) Some justification is required as to the focus on KV 7.2/7.3 and the use of KV 1.3 as a control. Why choose KV 1.3 particularly the authors need to justify the lack of screening against other potassium channels when talking about selected activation. There is a short description of other KV channels tested on lines 195-196 but this should be made more prominent and perhaps extend to a wider range of other ion channels.

Kv1.3 was not used as a control, but rather as a counter-screen to rule out extracts working via tannic acid, because we wanted to discover a novel Kv7.2/7.3 opener. We have now added more description of the basis for using Kv1.3 inhibition as a novel screening tool, a counter-screen to eliminate extracts opening Kv7.2/7.3 via tannic acid (final paragraph, page 4) to find instead novel Kv7.2/7.3 openers. We tested 9 other channel types and found effects on none of them, including all known Kv7 channel types (Figure 5). This makes gentisic acid the most selective known Kv7.3 channel opener. For the sake of historical comparison, the pan-neuronal-Kv7 channel opener and anticonvulsant retigabine was first reported in 1995, and was touted as Kv7 channel-selective for many years. In 2016, after 21 years, >400 publications (at that time; >700 now) and years of clinical use, another group found that at clinically relevant levels, retigabine inhibits another major neuronal channel, Kv2.1 (PMID: 27734968). That paper

has gone largely ignored and retigabine is often still referred to as Kv7-selective. If one checks the literature, few if any groups test as many Kv7 isoform homomer and heteromer combinations as we do when we profile Kv7 openers; we test them all. In addition, In the present manuscript, we already show that not only does GA lack effects on other Kv7 channels, it also does not affect Kv2.1, Kv1.1, Kv1.2 and Kv1.3. We have also now added a new figure (Figure 8) showing chemical selectivity and its basis as well (out of 12 gentisic acid analogs, only one shows any effect on Kv7.3). We have nevertheless added a statement in the Discussion that future research will include an even broader specificity screen (page 14, paragraph 1), but GA is the most Kv7.3 isoform-selective compound currently known. We have also added a fuller description for the reasoning behind Kv7.2/7.3 as a target (final paragraph, page 3).

4) Line 105 - justify the exclusion criteria described here

We now include the justification (page 5).

5) Line 115-116 - this seems rather vague account here.

We are not sure what is meant by “vague account” for this section. We had several botanists from Santa Monica Mountains National Recreation Area view the sugar bush hybrid and they indicated there are garden escape hybrids in the wash from which we collected the plant that defy definitive identification. As this plant was not the main focus of the study we did not attempt further identification, which likely would have been fruitless in any case.

6) The legend for Figure 1 refers to a previous figure in reference 25. However reference 25 has not been published anywhere so it is impossible to compare the earlier version of the figure with the one shown here.

We included that manuscript as an addendum for the reviewers in the previous submission and we include it again in this resubmission. It is now in press.

7) Lines 127-134 are really a description of the methods (or rather the justification for the methods chosen) for quantification and, as such, should probably sit in the method section

We understand the comment but we have found that especially for a general audience it is more effective to walk readers through the reasoning behind the voltage protocol while they view the results, so we prefer to keep that section in the Results.

8) Figure 2 could be rearranged to show the most important effect first namely the effect of Portia tree extra as this is a subject or the focus of the rest of the paper - this might be more effective than putting a box around these data. This is, after all, the key finding of the paper.

The order was purposeful so that we start at the top with the greatest Kv1.3 inhibitory effects and proceed to the least Kv1.3 inhibitory effects, i.e., the same order as in Figure 1. This helps with the order of text as we explain how we ruled out extracts with tannic acid as the active compound, as we proceed with describing the effects in the Results section. We have now added an explanation of the order to clarify this (page 6, paragraph 2).

9) There are some anomalies in the data in figure 3. The control resting membrane potential varies quite considerably from one set of experiments to another so that although gentisic acid

hyperpolarises the membrane potential by about 10 millivolts this really only takes it to the resting value seen in the experiments with coumarin for example. One could argue that in these coumarin experiments the membrane potential is already hyperpolarized before the addition of the drug and therefore the compound would not be able to hyperpolarize further. Despite this, the effect of gentisic seems clear enough to warrant further study as described subsequently in the paper.

This natural variation in oocyte E_m between different sets of oocyte batches does not in our experience dictate the ultimate shift magnitude caused by openers, i.e., a compound that shifts E_m by -15 mV will still do that whether the oocyte starts at -50 or -70 mV.

10) I don't think the authors make enough of the potency of gentisic acid. In many experiments 10-100 μ M concentrations are used, yet figure 4 d,e show that 10 nM is almost maximally effective. Comparing 10 nM gentisic acid on KV7.3 with 10s of μ M on other channels would be extremely persuasive.

We thank the reviewer for noting the high potency of gentisic acid! We do have that comparison in Figure 4, as directly under Figure 4d (10 nM exemplar for Kv7.2/7.3), there is the lack of effect of 100 μ M gentisic acid on Kv7.2 (Figure 4h). We have now accentuated that difference by adding blow-ups of all the tail currents in Figure 4 (in blue boxes).

11) The raw data traces show examples of beautiful current recordings for which the authors should be commended. However, the effects of compounds are measured/quantified on steps back to -30 mV following the test steps which are only a small fraction of the traces shown. These steps to -30 mV bring out the effects of the compounds most clearly but are perhaps lost in the raw data traces. Could this section of the traces be magnified with, perhaps, the whole trace shown as insets to bring out the effects of compounds?

Great suggestion - we have now blown up all the tail currents in Figure 4 (insets in blue boxes) and also traces for the Portia Tree effect shown in Figure 2 (insets in blue boxes), to emphasize the region from which the tail current graphs are quantified.

12) There is a bell shaped effect of gentisic acid seen on seizures. Why do the authors think this is so? How do the concentrations used here in mg/kg relate to the concentrations used in the electrophysiological experiments. The latter suggests that really low concentrations (10 nM) have a large effect and that whilst there is a saturation of the effect on KV7.3 containing channels there is not a reversal at high concentrations.

The 2mg/kg dose is equivalent to 13 μ M gentisic acid, although the concentration at the target in vivo is obviously expected to be much lower. We speculated in the Discussion that there may be a lower-affinity target, which is not even necessarily an ion channel, interaction of which with gentisic acid might interfere with the anticonvulsant effects, but we do not have the answer to this question, and it is likely to be complex. We have added additional discussion (page 14, paragraph 1).

13) Lines 305-324. Perhaps the authors could speculate further as to why the compound is so selective for KV7.3 containing channels and why KV7.2/7.3 channels are more responsive in terms of transducing the effect of binding into channel opening.

We have good evidence for another compound (carnosic acid) that the Kv7.3 voltage sensor is more susceptible to opening via some small molecules binding in the pocket between the pore and voltage sensor, than is Kv7.2. We are currently preparing a manuscript on that. We can also examine this for gentisic acid in a future study; it requires studying a large panel of alanine-scanning mutagenesis mutants and also MD simulations. We have added more speculation as requested in the current Discussion (page 15, paragraph 1).

Reviewer #2 (Remarks to the Author):

The paper by Abbott and Manville describes the identification of gentisic acid (GA) as a novel and potent Kv7.2/3 channel activator present in the flower/leaf extract of portia tree (*Thespesia populnea*). GA was identified following an extensive high throughput screen (thallium flux assay) of 1444 plant extracts and follow up validation testing on Kv7.2/3 channels expressed in *Xenopus oocytes*. Protocatechuic acid (PA) was also identified yet had lesser effects compared to GA. The screening protocol importantly excluded plant extracts with known high tannic acid levels, which itself has Kv7.2/3 channel activity that would otherwise occlude the ability to identify novel compounds.

The search for potent and selective Kv7.2/3 channel openers are of clinical interest for use as anticonvulsant agents in the treatment of epilepsies and seizure disorders. GA demonstrated high potency, with EC50's of 2-15 nM for different channel effects, and also demonstrated high selectivity with little to no effect on other Kv channels tested. Mechanistically, evidence is provided the GA binds to W265 in the Kv7.3 S5 segment via pi-stacking, where the W265L mutation resulted in a complete loss of GA-induced channel activity. This site has been previously implicated in the binding and actions of retigabine.

The experimental findings of GA actions on Kv7.2/3 channels is compelling and will be of general widespread interest. The experimental findings of GA actions as an anticonvulsant in the mouse seizure model, however, were less convincing. Pre-administration of GA (i.p. injection) was found to delay the acute onset of seizures induced by pentylenetetrazole (PTZ) injected 30 minutes later in mice. This effect, however, was only statistically produced at one dose tested (2 mg/kg), where higher doses (10 or 20 mg/kg) did not produce the same effect. From the data presented in figure 6, most mice did not respond at the 2 mg/kg dosage, where a small subgroup of 'responders' demonstrated a latency greater than the control latency value (~150 s). Thus, the acute anticonvulsant activity of GA that is presented, is currently the weakest part of the paper where the claim GA "ameliorates pentylenetetrazole-induced seizures in mice" is not overly convincing. Additional experiments or further discussion with possible explanations would help. Nevertheless, a more comprehensive testing of GA antiseizure activity (acute and chronic) is undoubtedly anticipated and expected to be the subject of follow up pre-clinical studies. The discussion of the portia tree being used as a Polynesian traditional anticonvulsant is quite interesting given the findings of GA and PA on Kv7.2/3 channels.

We thank Reviewer 2 for the highly positive discussion. We agree that the more comprehensive testing of GA antiseizure activity (acute and chronic) can be the subject of follow up pre-clinical studies. The 2 mg/kg dose equates to 13 μ M. We have added further discussion about the PTZ study findings at page 14, paragraph 1.

Minor

Delete line 848 in the figure legend for figure 6.

Done

Reviewer #3 (Remarks to the Author):

This report extends the work of this group in identifying new modulators of potassium channel action in extract from plants collected in California and the US Virgin Islands. In this report, their focus is on activators of the voltage-gated potassium Kv7.2/3, which is a target for anticonvulsant drug development. They included a counter-screen against Kv1.3 to select against inhibitors of this channel. Seven extracts were identified that increased Kv7.2/3 thallium flux by 76-150%, with Kv1.3 inhibition <93%.

As a portia tree extract showed a promising profile, the effects of 12 of the most abundant compounds present in this extract, as identified by others, were investigated. Gentisic acid (GA) showed an EC50 for increasing Kv7.2/3 current at -60 mV of 15.6 nM, and EC50 values for shifting V0.5act and EM of Kv7.2/3 of 2.8 and 2.4 nM, respectively. GA did not affect Kv7.2 homomers but did open Kv7.3* channels.

It was then shown that GA ameliorated pentylenetetrazole-induced seizures in mice. GA at 2 mg/kg more than doubled the latency to first seizure compared to mice treated with vehicle alone, although increasing concentrations of GA had weaker effects on latency. As acknowledged in the Discussion, this would need to be thoroughly understood to optimize dosage and avoid potential proconvulsant effects before GA could be considered for use as an anticonvulsant.

Others have proposed that retigabine interacts with S5 W265 on Kv7.3 or its equivalent on Kv7.2 (W236). Docking conducted in this study suggests that GA could also interact with Kv7.3-W265. Mutation of Kv7.3-W265 to Leu eliminated the effects of GA (100 μM) on Kv7.2/3 channels, as did mutation to Leu of both Kv7.3-W265 and Kv7.2-W236. It is remarkable that such limited interactions between GA and the channel could confer selectivity. As indicated in the Discussion, there seems to be something of a disconnect between binding selectivity and functional selectivity in determining small molecule Kv7 opener selectivity, although the molecular basis for this is not well understood.

While this study is certainly of interest, I question whether it is appropriate for a chemistry journal, as opposed to one focusing on, for example, pharmacognosy or molecular pharmacology. There is essentially no chemistry in the current manuscript.

In response to the comment about chemistry, we were inspired to conduct additional studies, replacing the prior Supplementary Figure 1 with a more comprehensive SAR, presented as a new Figure 8, in which we show the results from testing 12 gentisic acid analogs on Kv7.2/7.3. In addition to now having a better understanding of the chemical requirements for GA-based Kv7.3 openers, we found that one of the analogs (the FDA-approved vasoprotective drug, calcium dobesilate) exerts opening effects on Kv7.2/7.3 channels similar to those of GA. We describe the SAR in the context of the chemical properties of the compounds, in the Results section (final paragraph of the Results section) and the Discussion (page 13, final paragraph). We also added a line in the abstract about the calcium dobesilate discovery.

In response to the comment about selectivity, we have had several papers published on the basis for selectivity of Kv7 channel openers that we have discovered to be

functionally but not physically selective, e.g., GABA, aloperine, carnosic acid. We are among the very few groups to have actually investigated physical binding of compounds to potassium channels; most groups infer everything about selectivity from functional studies, but as our prior research has shown, this can be very misleading. We have found that many compounds bind to Kv7 channels without functional consequences; e.g., we found that the neurotransmitter GABA physically binds to Kv7.2, 3, 4 and 5 (but not Kv7.1, which lacks the S5 tryptophan) but only opens Kv7.3 and Kv7.5 (PMID: 29748663). We are in the process of a major study of this phenomenon for carnosic acid, which we found to open Kv7.3 but not Kv7.2, similar to GA. The work involves alanine scanning mutagenesis and MD simulations and would be outside the scope of the present study because of the size of the undertaking, but we plan to perform a similar study with GA once the carnosic acid paper is complete.

Line 246-247: 'T. populnea wood is used to construct furniture, musical instruments, kitchenware
247 and canoe paddles, etc., while the bark makes effective rope and caulk.' This is irrelevant and should be deleted. The same applies to lines 334-340 in the Discussion.

Removed.

Data Availability: Source data for Figures 1-7 and Supplementary Figure 1 are available at Dryad data repository: xxxxxxx. The repository information is required.

We have added the repository information: <https://doi.org/10.5061/dryad.547d7wmhn>.